# Gene Therapy for Lysosomal Storage Disorders: Ongoing Studies and Clinical Development

**DOI:** 10.3390/biom11040611

**Published:** 2021-04-20

**Authors:** Giulia Massaro, Amy F. Geard, Wenfei Liu, Oliver Coombe-Tennant, Simon N. Waddington, Julien Baruteau, Paul Gissen, Ahad A. Rahim

**Affiliations:** 1UCL School of Pharmacy, University College London, London WC1N 1AX, UK; amy.geard.16@ucl.ac.uk (A.F.G.); wenfei.liu.10@ucl.ac.uk (W.L.); o.coombe-tennant@ucl.ac.uk (O.C.-T.); a.rahim@ucl.ac.uk (A.A.R.); 2Wits/SAMRC Antiviral Gene Therapy Research Unit, Faculty of Health Sciences, University of the Witwatersrand, Johannesburg 2193, South Africa; s.waddington@ucl.ac.uk; 3Gene Transfer Technology Group, EGA Institute for Women’s Health, University College London, London WC1E 6HX, UK; 4Metabolic Medicine Department, Great Ormond Street Hospital for Children NHS Foundation Trust, London WC1N 1EH, UK; j.baruteau@ucl.ac.uk; 5Great Ormond Street Hospital Biomedical Research Centre, Great Ormond Street Institute of Child Health, National Institute of Health Research, University College London, London WC1N 1EH, UK; p.gissen@ucl.ac.uk

**Keywords:** lysosomal diseases, gene therapy, viral vectors

## Abstract

Rare monogenic disorders such as lysosomal diseases have been at the forefront in the development of novel treatments where therapeutic options are either limited or unavailable. The increasing number of successful pre-clinical and clinical studies in the last decade demonstrates that gene therapy represents a feasible option to address the unmet medical need of these patients. This article provides a comprehensive overview of the current state of the field, reviewing the most used viral gene delivery vectors in the context of lysosomal storage disorders, a selection of relevant pre-clinical studies and ongoing clinical trials within recent years.

## 1. Introduction

Lysosomal storage disorders (LSDs) form a group of genetic errors of metabolism, comprising more than 70 different diseases [1]. These monogenic disorders result from defects in proteins crucial for lysosomal function, such as lysosomal hydrolases, as well as transporters, integral membrane proteins, co-factors and enzyme modifiers or activators [2]. Moreover, mutations in non-lysosomal proteins that play a role in lysosome-related processes can also cause LSDs. The lysosome is a complex organelle, involved in several cellular processes such as autophagy, signalling cascades, lipid and calcium homeostasis, exocytosis, membrane repair and cell viability [3,4]. As a consequence of genetic defects, these cellular functions and mechanisms are disrupted, leading to inflammatory responses and ultimately cell death. Typical LSDs involve accumulation of undegraded substrates in the lysosome due to defective hydrolase enzymes, and classification of the disorder can be based on the nature of the accumulated substrate. However, the identification of additional defects in integral proteins for which the function is not fully understood has broadened the traditional classification of LSDs [5]. The disease can also be classified by the causative underlying defect. Although LSDs are individually considered rare diseases, some populations have a higher carrier frequency, and the collective prevalence as a group is relatively common, with an estimated incidence of 1 in 5000 live births [1].

Specific treatments that aim to correct the metabolic defect have been developed for many LSDs. These include haematopoietic stem cell transplantation (HSCT), enzyme replacement therapy (ERT), pharmacological chaperone therapy (PCT) and substrate reduction therapy (SRT). HSCT involves the transplantation of haematopoietic stem cells, usually derived from bone marrow. The rationale relies on transplanted stem cells engrafting in the bone marrow and differentiating into various cells of the haematopoietic lineage; some of the cells then cross the blood brain barrier (BBB) and potentially may differentiate into resident microglia of the central nervous system that can secrete functioning lysosomal enzymes [6,7]. HSCT is considered the standard or optional therapy for some LSDs such as mucopolysaccharidosis type I (MPS I), when diagnosed before 2.5 years of age [8], alpha mannosidosis and globoid cell leukodystrophy, where early-stage treatment correlates with better clinical outcomes, and it could be indicated for other inherited metabolic disorders [9]. However, this approach is not feasible in disorders such as the late infantile form of metachromatic leukodystrophy, where engraftment and expansion of donor-derived cells are significantly slower than the rapid progression of the disease, and the treatment is not able to prevent the development of neuropathology [10].

ERT is based on the premise that LSDs could be treated by administering the recombinant functional enzyme to patients to compensate for their abnormal or defective protein. The concept was originally suggested by Christian de Duve in 1964 [11] and is supported by the discovery that many lysosomal proteins are targeted via the mannose-6-phosphate (M6P) receptor pathway [12]. According to the mechanism of ‘cross-correction’, the addition of a mannose group to the recombinant enzyme facilitates interaction with the mannose receptor on the surface of macrophages, allowing uptake by neighbouring cells and facilitating its transport to the lysosomes [13]. Alglucerase was the first recombinant enzyme approved for use in humans for the treatment of type 1 Gaucher disease in 1991 [14]. The enzyme was originally harvested from human placental tissue; however, subsequent therapies producing recombinant enzyme using gene activation technology in Chinese hamster ovary cells (Imiglucerase, Genzyme), human fibrosarcoma cells (Velaglucerase alfa, Takeda) or a carrot cell line (Taliglucerase alfa, Pfizer) are equally efficacious [15,16,17]. ERT is now considered the standard of care for many LSDs and is currently approved for the treatment of Fabry, Gaucher and Pompe diseases, alpha-mannosidosis, acid lipase deficiency, late infantile neuronal ceroid lipofuscinosis type 2 (CLN2) and mucopolysaccharidoses type I, II, IVA, VI and VII [18,19]. ERT is typically given by repeated intravenous (IV) administrations, but despite being effective, the regular nature of administration affects the quality of life of patients, as well as having a considerable economic impact on the health system [20]. In addition, repeated infusions of high concentrations of the recombinant enzyme can lead to the development of immune reactions to the protein requiring complex management [21,22]. Another drawback of this therapy is that it is unable to cross the BBB, therefore preventing treatment of the neurological phenotypes of some LSDs [19]. To overcome the limitations of using ERT to treat neuropathology, direct administration to the cerebral ventricles is successfully used to deliver the tripeptidyl peptidase 1 enzyme (cerliponase alfa, Brineura^®^) to CLN2 patients [23]. The nature of the affected tissue limits the use of ERT for certain LSDs, as some tissues can be refractive to penetration by most molecules [19], such as skeletal manifestations of Gaucher disease, cardiac valves in MPS or the neurological phenotype of many LSDs.

PCT involves the use of molecular chaperones that may stabilise conformation of unstable or misfolded proteins due to mutations that cause LSDs. This prevents rapid degradation of proteins and allows them to perform their enzymatic function [24]. The only approved therapy is Migalastat for the treatment of Fabry disease [25,26]; however, Ambroxol is currently in a phase 2 clinical trial (NCT03950050) as a potential therapeutic for type 1 Gaucher disease. The PCT approach shows promise, but its use is limited to mutations that affect only the stability of the mutant enzyme and may not be effective in those cases where the mutation affects the catalytic site of the protein. Endogenous molecular chaperone proteins can be activated in response to the heat shock response (HSR) following cellular stress or damage [27]. Heat Shock Protein 70 (HSP70) activators, such as the small molecule Arimoclomol (Orphazyme), can induce the chaperone response, promoting correct protein folding or clearance of damaged proteins in the cytosol. Arimoclomol is currently being investigated for the treatment of Niemann-Pick type C (NCT02612129) and Gaucher patients (NCT03746587).

SRT has been investigated as a potential therapy to treat neurological phenotypes. These small molecules aim to reduce accumulation of undegraded substrate by limiting its synthesis or the synthesis of its precursors [28,29]. Miglustat (Zavesca, Actelion Pharmaceuticals) and Eliglustat (Cerdelga, Sanofi Genzyme) have been approved for the treatment of type 1 Gaucher disease as second- and first-line therapies, respectively [30,31]. However, Miglustat has not been proven to be more efficacious than ERT [32]. Eliglustat is unable to cross the BBB and therefore cannot be used to treat neuronopathic forms of the disease. Miglustat has also been approved by the European Medicines Agency for the treatment of Niemann-Pick type C (NPC) as it showed a significant reduction in neurological disease progression [33].

Gene therapy is the delivery of nucleic acids to cells for patient benefit using viral or non-viral derived vectors. In the context of LSDs, this is usually delivery of a functional copy of the defective gene. LSDs are an ideal candidate disease group for gene therapy because they are typically well-characterised monogenic disorders for which the defective gene has been identified and animal models have been developed that allow for preclinical testing. For some disorders, the newly synthesized enzyme is secreted into systemic circulation for recapture and use by non-neighbouring cells. This ‘cross-correction’ process allows certain organs, such as the liver, to act as an enzyme factory where the protein is produced and delivered to other organs via blood circulation. Moreover, the amount of enzyme correction required to sufficiently clear the accumulating substrate is estimated to be between 1 and 10% of normal concentrations, although this will vary by disease [34]. Thus, a complete recovery of the physiological enzyme activity might not be needed to ameliorate disease phenotypes. However, those LSDs that are caused by defective membrane bound proteins that are not secreted are more challenging since there is no cross-correction. Ideal viral vector systems exploit the efficient ability of viruses to infect cells while preventing the expression of viral genes that can lead to replication and subsequent toxicity. Importantly, genes encoding replication and capsid or envelope functions are removed from the viral genome and provided in trans during the vector production process. Viral genes are replaced with the expression cassette sequences encoding the transgene of interest, promoter of choice and other regulatory elements. Great focus has been placed on vectors that avoid activation of oncogenes if nearby vector integration occurs and can control gene expression to target cells in order to prevent toxicity and the efficient gene transfer into these target cells, whether ex vivo or in vivo [35]. The most common viral vector systems used include retroviruses, lentiviruses, adenoviruses and adeno-associated viruses. Each of the different viral vector systems possesses certain characteristics that determine its suitability for specific applications (Table 1, Figure 1).

### 1.1. Retroviral Vectors

Retroviruses are lipid-enveloped particles that contain a RNA genome. The viral envelope glycoprotein is responsible for interaction of the retroviral particles with receptors on target cells, thus controlling the host tropism and biodistribution [36]. Substitution of the receptor-binding proteins on the viral envelope with those from other, unrelated viral strains is known as pseudotyping [37]. This engineering strategy has been successfully used to expand the target range of retroviral vectors. Following entry into target cells, the RNA genome is retrotranscribed by reverse transcriptase into double-stranded linear DNA strands that then randomly integrate into the host genome [36]. While integration into the host genome enables long-term gene expression in dividing cells, possible insertion of genetic material near oncogenes could cause disease [38]. Moreover, further undesirable traits of retroviral vectors include their highly immunogenic nature and their ability to only transduce dividing cells [37]. Therefore, they are most commonly used for ex vivo applications.

### 1.2. Lentiviral Vectors

Lentiviruses are a genus of the retrovirus family. Lentivirus-based vectors mediate more stable transgene expression and efficient gene transfer than previous retroviral vectors [35]. The lentiviral genome has been engineered to contain self-inactivating long terminal repeats (LTRs), and the vectors have been shown to significantly reduce insertional genotoxicity [39,40,41]. The current generation of lentiviral vectors is based on the HIV type 1 virus, elements of which integrate into the host cell genome, posing a risk for clinical translation. Recently Vink and colleagues [42] described a novel lentiviral vector with reduced packaging of HIV-1 sequences within the LTRs. LTR1 vectors showed increased safety profile, titre and transgene expression. These vectors are widely used in the context of ex vivo gene therapy but have also been recently employed in in vivo trials where, when administered to the patient, they are able to give rise to robust and stable transgene expression [43,44]. Unlike retroviral vectors, such as gammaretroviral vectors, lentiviral vectors are able to transduce non-dividing cells by nuclear targeting using the nuclear import machinery of the host cell [45]. Moreover, these vectors do not usually induce a severe immune response upon administration. However, it is possible for T-cell responses to be activated against the transgene delivered by the vector. Lentiviral vectors pseudotyped with the G glycoprotein of the vesicular stomatitis virus (VSV-G) exhibit expanded tropism compared with the original HIV1-derived lentivirus [46]. Pseudotyping involves substituting the receptors of the original envelope with those from VSV-G, which may also improve the stability of the vector [46]. This vector has been successfully used, preclinically, in both in vivo and ex vivo applications. Lentivirus vectors have been used with resounding success clinically in ex vivo gene therapy, for example, in the treatment of primary immune deficiencies [47] and as an anti-cancer therapy in the generation of CAR T-cells [48].

### 1.3. Adenoviral Vectors

Adenoviruses are non-enveloped viruses with an icosahedral capsid containing a double-stranded DNA (dsDNA) genome [37]. More than 50 adenoviral serotypes derived from human and other species have been identified, and while the natural viruses cause disease in humans, they can be engineered and exploited to safely and efficiently deliver genes to a variety of cell types [36]. Unlike with retroviral vectors, a packaging cell line has not been engineered. To increase the carrying capacity of the vector, a helper-dependent vector system was developed [49], where a helper virus contains the genes needed for replication, thus allowing 28–32 kb of packaging capacity contained between two inverted terminal repeat sequences (ITRs) in a second vector. ITRs play a role in adenoviral DNA replication and transcriptional activity. This second vector also contains the normal packaging recognition signal, allowing its packaging and release from cells. Adenoviral vectors are considered to be among the most immunogenic of the viral vector groups, which limits their use for treatment of monogenic disease for in vivo gene therapy [50]. However, this makes them an excellent choice for use in oncolytic virotherapy [51] and as a vaccine against, for example, Severe Acute Respiratory Syndrome Coronavirus 2 SARS-Cov2 [52].

### 1.4. Adeno-Associated Viral Vectors

Adeno-associated viruses (AAVs) are part of the Parvoviridae family and are replication-deficient non-enveloped viruses. AAVs typically require a helper virus, such as the adenovirus, to facilitate infection. In the absence of a helper virus, the AAV genome remains latent in the host cell [53] in an episomal state. AAVs are not known to cause disease in humans, with 11 known human viral serotypes presenting wide and differential cellular tropism [54]. AAV2 is the most well-known serotype, and current AAV vectors are based on its genome. The wild-type AAV genome contains rep and cap genes that encode proteins involved in DNA replication [55] and capsid formation [56], respectively. The genome is flanked by 145-bp inverted terminal repeats (ITRs) which function in cis as packaging signals, denoting the coding region that will be packaged into the capsid and inserted into the host cell [57]. In recombinant AAV (rAAV) vectors, the rep and cap genes are provided in trans and are replaced with the transgene cassette that is flanked by the ITRs. Using this method, the recombinant AAV2 genome has successfully been packaged into various capsids of other serotypes, thus enhancing the tropism of this vector [58,59]. The rAAV system is the most promising for low-immunogenic [60], long-term gene expression [61,62] and is able to transduce a wide range of tissues. Similar to the lentivirus, wild-type AAV serotypes are not highly immunogenic. However, T-cell responses to the transgene may arise after administration of the AAV-based vector, particularly if the vector transduces cells that are involved in antigen presentation, such as dendritic cells [37,63]. It is important to note that majority of the human population has been exposed to wild-type AAV serotypes, and the prevalence of neutralizing anti-capsid antibodies is high [64,65]. Moreover, the development of neutralizing antibodies after the first administration in patients may hinder vector re-administration [66].

## 2. In Vitro Haematopoietic Stem Cell Gene Therapy

The theoretical basis of haematopoietic stem cell transplantation for LSDs lies in (1) identification of the M6P receptor pathway [67], through which lysosomal enzymes secreted by a cell can bind to the M6P receptor on the cell membrane of surrounding cells with subsequent uptake and transport to the lysosome and (2) transplanted haematopoietic stem cells (HSCs) or their progeny can settle in various tissues to contribute to the local macrophage populations and thus become sources of the required lysosomal enzyme.

The history of HSCT for LSDs started from conventional allogeneic HSCT, which employs HSCs from a healthy person to supply the lysosomal enzyme deficient in the transplant recipient. In the field of LSDs, conventional allogeneic HSCT has performed well in treating MPS I-H, which is caused by *IDUA* mutations resulting in alpha-L-iduronidase deficiency and affects multiple tissues including the central nervous system (CNS). The first trial was reported in 1981, which described biochemical and clinical improvements of an MPS I-H patient that received allogeneic HSCT at one year of age [68]. Since then, more HSCT has been performed for the treatment of MPS I-H, but the clinical outcomes are highly variable among patients as well as among different tissues, as reviewed by Aldenhoven et al. in 2008 [69]. More recently, a long-term longitudinal study including 217 patients from multiple clinical centres was reported to provide more information on the efficacy of allogeneic HSCT for treating MPS I-H [70]. Encouragingly, dramatic improvement of clinical course and lifespan was observed. However, a significant residual disease burden remained, and as lifespan was increased, various clinical manifestations emerged during long-term follow-up. In general, the outcomes of HSCT are affected by various aspects of the transplant such as haploidentity between the donor and recipient, severity of myeloablative conditioning, phenotypic severity at the age of HSCT, degree and persistence of donor chimerism and post-transplant complications [71,72]. A critical predictor for better outcome is sufficient circulating enzyme concentrations obtained post-HSCT, which could be affected by the degree of donor chimerism. This is particularly challenging in the CNS, and limited therapeutic benefits for the CNS are a relatively common finding from LSD treatment with HSCT.

Haematopoietic stem cell gene therapy (HSC-GT) holds great potential. An advantage of HSC-GT is the ability to perform autologous transplantation, which bypasses the difficulty in donor selection for allogeneic HSCT and avoids the risk of graft-versus-host disease. Addition of gene therapy to HSCT using viral vectors may provide supraphysiological expression of the required lysosomal enzymes as with other gene therapies. This is of particular interest for LSDs, as secretion of the therapeutic protein is enhanced due to gene overexpression in the transplanted cells, resulting in a higher degree of cross-correction. Considering the heredity of all LSDs, matched sibling donors (commonly employed in allogeneic HSCT) may also bear the disease mutation and thus cannot provide HSCs of sufficient therapeutic efficacy. The general approach of HSC-GT proceeds as follows: (1) autologous HSCs are collected from the patient, (2) collected HSCs are genetically corrected by transduction with a viral vector harbouring the therapeutic gene and (3) the transduced HSCs are transplanted back into the patient following transplant conditioning treatment similar to that required for HSCT. The most commonly used vectors for HSC-GT are lentiviral and gammaretroviral vectors. Both are able to transduce HSCs as well as to integrate into the host genome, allowing permanent target gene expression by transduced cells and, more importantly, by their progeny in the context of HSCT. However, genome integration is always a double-edged sword with risk of oncogenesis and mutagenesis over time, as discussed above. Thus, gene therapy employing these integrating vectors requires careful safety studies, preferentially with large-scale analysis and long-term follow-up. To date, clinical trials of HSC-GT have been initiated for treatment of various LSDs, including metachromatic leukodystrophy (MLD), MPS IIIA, MPS I-H, cystinosis, Gaucher disease and Fabry disease (Table 2).

### 2.1. Metachromatic Leukodystrophy (MLD)

One of the best examples of HSC-GT for treating LSDs, to date, is for metachromatic leukodystrophy. MLD is caused by a deficiency of arylsulfatase A (ARSA) due to mutations in the *ARSA* gene. ARSA is a soluble lysosomal enzyme with its lysosomal targeting dependent on the M6P receptor pathway. The physiological properties of ARSA, together with the extensive neurological involvement of MLD, offer hope for the disease to be a potential candidate for HSC-GT. Preclinical HSC-GT studies provided supportive data for clinical trials. Initially, Matzner et al. investigated HSC-GT using a mouse model of MLD and bone-marrow-derived HSCs transduced with a retroviral vector (murine stem cell virus, MSCV) harbouring the human *ARSA* gene [73,74,75]. Significant correction of lysosomal storage was observed in visceral organs such as the liver and kidney. However, improvement of CNS pathology and behavioural performance was only minor, despite the forebrain ARSA concentration in the treated MLD mice being increased to 10–33% of that of the wild-type. This may suggest that, for MLD, a better therapeutic outcome in the CNS may require even higher expression of the therapeutic enzyme and/or more extensive CNS microglia/macrophage reconstitution by progeny of the transduced HSCs allowing for broader cross correction. Furthermore, the authors found that although a long-term high bone marrow engraftment rate was obtained, the transduction efficiency and the quantity of transgene expression in the HSCs were highly variable and thus hampered consistency of the enzymatic correction, which could be at least partially due to the integration patterns of retroviral vectors. A few years later, Biffi et al. tested advanced-generation lentiviral vector-mediated HSC-GT in MLD mice [76,77] and achieved strikingly enhanced outcomes. They showed that both neurological pathology and behavioural performance were vastly improved, whether transplanted at the pre-symptomatic [76] or early-symptomatic [77] stage. Such corrections were not achieved by using conventional HSCT from wild-type donors without transduction, suggesting the importance of overexpressing the therapeutic gene to boost efficacy. Additionally, using the HSC-GT expressing HA-tagged *ARSA* gene, they detected ARSA in both microglia/macrophages in the CNS (representing transduced mononuclear cells engrafted into the CNS) and neurons plus other cell types (providing evidence for cross correction), whereas HSC-GT expressing *EGFP* resulted in GFP detection only in microglia/macrophages in the CNS [77]. Compared with previous studies, the therapeutic success of this work, as suggested by the authors, may lie in the more efficient HSC transduction and transgene overexpression by advanced-generation lentiviral vectors to reach the threshold enzyme concentrations required for a good therapeutic effect. This threshold could vary between different LSDs and thus should be investigated on a case-to-case basis.

Based on these preclinical studies, clinical trials have been performed to treat MLD patients. Promising outcomes have been reported from a phase 1/2 clinical trial of HSC-GT in MLD children (NCT01560182) [78,79]. MLD patients with mutations associated with early-onset MLD received HSC-GT transduced with lentiviral vectors harbouring human *ARSA* at pre-symptomatic or very early symptomatic stages, with myeloablative busulfan pre-transplant conditioning. The patients were then followed up for 18–54 months post transplantation. Stable engraftment of transduced HSCs was observed during follow-up, and ARSA activity was reconstituted to physiological or supra-physiological concentrations in haematopoietic cells and in the cerebrospinal fluid (CSF). The therapeutic efficacy was promising. Eight of nine patients showed prevention of disease onset or arrest of progression in accordance with clinical and instrumental assessment. In addition, skin biopsy exhibited occurrence of peripheral nerve remyelination in some of the patients. No serious adverse effects due to the medicinal product were reported during follow-up, and no haematopoietic malignancy was observed. The most frequent adverse events were cytopenia and mucositis, which are common complications of the chemotherapy conditioning for HSCT. A recent case report by Calbi et al. showed post-HSCT endothelial injury could be prevented by using Defibrotide and adjusting busulfan exposure to a lower but still myeloablative area under the curve (AUC) [80], as shown for other HSCT indications. To date, HSC-GT has shown exciting outcomes for MLD, which offers hope and valuable information for development of HSC-GT for other LSDs. In late 2020, the European Commission granted full market authorisation for Libmeldy (Orchard Therapeutics) for the treatment of MLD patients.

### 2.2. Mucopolysaccharidosis Type IIIA (MPS IIIA)

Recently, a phase 1/2 clinical trial was initiated (NCT04201405) for treatment of mucopolysaccharidosis type IIIA (MPS IIIA) [81]. MPS IIIA is a neurodegenerative LSD, caused by deficiency in the lysosomal enzyme *N*-sulfoglucosamine sulfohydrolase (SGSH) due to mutations in the *SGSH* gene. Patients usually have onset of disease in their first decade of life and present with heparan sulphate accumulation, progressive neurodegeneration and behavioural abnormalities, leading to death in their second decade. Currently most therapies for MPS IIIA are supportive treatments directed towards multisystem complications, whereas no effective disease-modifying therapy is available to prevent disease progression [81]. HSC-GT was initially investigated in an MPS IIIA mouse model but did not display dramatic therapeutic benefits [82]. The authors compared HSC-GT using wild-type donor cells transduced ex vivo with lentiviral vectors expressing SGSH (LV-WT-HSCT) or MPS IIIA deficient cells transduced ex vivo with lentiviral vectors expressing SGSH (LV-IIIA-HSCT) versus conventional wild-type donor cell transplant (WT-HSCT). Among these three strategies, the LV-IIIA-HSCT is apparently the best representative of autologous HSC-GT in humans which, however, showed no significant effect on behavioural performance, despite high bone marrow engraftment and the brain enzymatic activity correction to 7% of wild-type concentrations. The same research group then hypothesized that specifically enhancing the transgene expression in the myeloid lineage could make autologous HSC-GT more efficient in the myeloid precursors that can engraft in the brain, thus achieving better therapeutic benefits. Therefore, they optimised the vector by transducing MPS IIIA HSCs with a novel lentiviral vector containing a codon-optimized SGSH gene under the myeloid-specific CD11b promoter (instead of the ubiquitous phosphoglycerate kinase (PGK) promoter) [83]. While it was not clear whether engrafting cells would retain microglia characteristics (e.g., at the transcriptional level), the outcomes were promising; brain lysosomal storage accumulation, neuroinflammation and abnormal open-field behaviour of the MPS IIIA mice were all corrected. In preparations for clinical trials, the authors performed a further study to evaluate the efficiency of the CD11b vector in transducing human HSCs, as well as a safety study [84]. The authors showed that the clinical grade Good Manufacturing Practice (GMP) lentiviral vector expressing *SGHG* under the CD11b promoter could effectively transduce human HSCs, which could also be effectively scaled up for a sufficient dose necessary for the clinic. HSC-GT with this vector in a humanised mouse model resulted in effective engraftment and biodistribution, with no vector shedding or transmission to germline cells. The vector genotoxicity assay showed low transformation potential, which was comparable to other lentiviral vectors currently used in the clinic. Orchard Therapeutics has initiated a phase 1/2 clinical trial for autologous HSC-GT in MPS IIIA patients using this vector (ClinicalTrials.gov Identifier: NCT04201405). In summary, compared with other HSC-GT using ubiquitous promoters, this example offers evidence for a strategy using a myeloid-specific promoter to drive transgene expression as an alternative option of (1) boosting expression of the transgene in myeloid cells that engraft and differentiate into microglia-like cells in the CNS, which would be especially useful for LSDs with extensive CNS involvement and (2) preventing potential toxicity associated with supraphysiological enzyme concentrations in the general circulation.

### 2.3. Mucopolysacchridosis Type I, Hurler Variant (MPS I-H)

Although allogeneic HSCT has been used as a treatment, HSC-GT has also been studied for MPS I-H. The importance of studying HSC-GT in MPS I-H is to eliminate the problem of histocompatibility in allogeneic HSCT and to potentially provide higher concentrations of the therapeutic enzyme by overexpression for this most severe type of MPS I. An early study using an MPS I-H mouse model firstly evaluated potential therapeutic effects of HSC-GT mediated by an MND retroviral vector expressing the human *IDUA* gene [85]. The tissue concentrations of alpha-L-iduronidase activity achieved by *IDUA*-transduced HSCs were variable between experiments, but a significant reduction of glycosaminoglycan accumulation and histological improvement were consistently observed in visceral organs, with the exception of the brain. A few years later, a study led by Biffi et al. tested HSC-GT mediated by lentiviral vectors and showed complete therapeutic correction in a mouse model [86]. In particular, neurological and skeletal impairments, a challenge to conventional HSCT and other therapies, were successfully corrected. This example, as well as the HSC-GT studies in MLD, suggests that second-generation lentiviral vectors perhaps represent a more efficacious strategy than RVs in HSC-GT, especially for LSDs with CNS involvement. Additionally, with regard to the safety issue, lentiviral vectors show a lower risk of genotoxicity than retroviral vectors when used to transduce HSCs [39,87,88]. A phase 1/2 clinical trial was started in 2018 to evaluate the efficacy and safety of lentiviral vector-mediated autologous HSC-GT in the treatment of MPS I-H patients (ClinicalTrials.gov Identifier: NCT03488394). In early 2021, Orchard Therapeutics shared the first encouraging preliminary clinical data from the follow up of the initial proof-of concept study, showing sustained IDUA expression in the CSF and normalization of pathology biomarkers in all treated patients [89].

### 2.4. Gaucher Disease

Numerous and successful results in preclinical studies using a type 1 mouse model [90,91,92] demonstrated the feasibility and safety of HSC-GT for Gaucher patients. Gene therapy not only prevented disease development but resulted also in long-term amelioration of hepatosplenomegaly and blood parameters, correcting the already established phenotypes. Avrobio has initiated a phase 1/2 HSC-GT clinical trial (NCT04145037) where autologous CD34+ enriched HSCs are modified with a lentiviral vector and administered in conjunction with a conditioning regime to type 1 Gaucher subjects. The self-inactivating vector expresses a codon-optimised *GBA* sequence under the control of the elongation factor 1α short (EFS) promoter. The data on the first treated patient reported a significant decrease in the toxic metabolite Lyso-Gb1 and in the biomarker chitotriosidase, with consequent interruption of ERT [93]. Importantly this approach has the potential to improve bone mass and density; in fact, the skeletal system is usually not targeted by ERT, and bone crises are a common feature in the lives of Gaucher patients.

### 2.5. Cystinosis

Although the mechanism of ‘cross-correction’ is applicable for soluble enzymes, diseases caused by defects in transmembrane proteins can also be targeted with an ex vivo gene therapy approach. This is the case for cystinosis, a disease caused by mutations in the *CTNS* gene encoding the lysosomal membrane-specific transporter for cystine. Cystine accumulation appears in all tissues, and the major cause of death is severe renal damage. A preclinical study by Harrison et al. demonstrated in a *Ctns* deficient mouse model that HSCs transduced with a lentiviral vector expressing human *CTNS* were capable of achieving engraftment and expressing transgene in all tissues tested [94]. Cystine accumulation in tissues was largely reduced, including in the kidney. Renal function was assessed eight months post-transplantation, and in the treated *Ctns* deficient mice, serum creatinine, urine phosphate and urine volume were significantly decreased compared with non-treated *Ctns* deficient mice. To date, a phase 1/2 clinical trial has been initiated to evaluate the efficacy and safety of HSC-GT in cystinosis patients (ClinicalTrials.gov Identifier: NCT03897361).

### 2.6. Other Pre-Clinical Studies

In addition to MPS IIIA and I-H discussed above, HSC-GT has been studied in animal models for treating other MPS types such as MPS II [95,96,97], MPS IIIB [98,99] and MPS VII [100,101]. All these disorders involve the CNS, which likely makes them better candidates for HSC-GT because of the CNS engraftment of the enzyme-proficient donor HSCs. Although it has been revealed that the renewal and maintenance of microglia and other CNS myeloid cells are independent of circulating monocytes under physiological conditions [102], use of a myeloablating conditioning regimen prior to transplantation allows re-population of CNS myeloid cells with the donor immigrants [103], which then becomes the local source for the CNS enzyme correction. The majority of these studies utilised lentiviral vectors to transduce HSCs and showed therapeutic benefits, suggesting that HSC-GT appears to be a promising approach to treat LSDs with CNS involvement.

Numerous pre-clinical studies of HSC-GT have been performed in Pompe disease [104,105,106,107]. Pompe disease is caused by a deficiency of the acid alpha-glucosidase (GAA) enzyme due to mutations in the *GAA* gene, resulting in the accumulation of lysosomal glycogen in a variety of cell types. The typical cause of death in this disease is cardio-respiratory failure which results from hypertrophic cardiomyopathy and skeletal muscle weakness. An early study in the Netherlands [107] using a Pompe mouse model showed that HSC-GT with a lentiviral vector expressing *GAA* reversed cardiac remodelling and improved disease manifestations, without obvious toxicity caused by *GAA* overexpression. However, complete correction of lysosomal glycogen accumulation in the skeletal muscle and the CNS cells was not achieved, and motor performance was only partially improved. Recently, the same research group conducted an HSC-GT study for Pompe mice using a novel lentiviral vector expressing codon-optimised *GAA*, which resulted in significantly higher GAA activity in peripheral blood [106]. The treatment led to almost complete correction of glycogen accumulation in the heart, skeletal muscles, brain and other organs, as well as restoration of locomotor functions tested at 10–12 months post-transplantation. Furthermore, vector integration site analysis was performed and confirmed that none of the genes near common integration sites was an oncogene. These promising results appear to pave the way for clinical trials using this vector to treat Pompe disease.

### 2.7. Innovative Techniques

One of the major advantages of HSC-GT for LSDs compared to conventional HSCT is the ability to overexpress the therapeutic gene in the HSCs and their progeny. This results in higher enzyme concentrations and better cross-correction once the transgene-expressing cells repopulate various tissue compartments. HSC-GT is attractive for treating the neurological manifestation of the diseases compared to other therapies such as ERT that cannot cross the BBB and conventional HSCT which produces relatively lower enzyme levels in the CNS. As discussed above, obtaining sufficient tissue enzyme activity for clinical benefits is an important issue for some LSDs. This has been studied with various optimisiation methods, such as the careful selection of vectors and promoters and codon-optimisation of the gene sequence. There are also alternative approaches that have been studied in the field of LSDs. Gleitz et al. [95] investigated a lentiviral vector expressing the target gene fused with *ApoEII* for treating MPS II in animal models, which resulted in the human IDS protein fused to a receptor-binding domain of human ApoE. The theoretical basis of this experiment was the ability of the receptor-binding portion of ApoE to form a high-affinity binding complex with heparan sulphate, which could potentially promote cross-correction in addition to uptake via the M6P-R pathway. The plasma enzyme concentration and uptake across the BBB were elevated, whereas the enzymatic activity was not affected by this modification. When compared to HSCT without vector transduction and HSC-GT with a lentiviral vector expressing normal IDS in an MPS II mouse model, this IDS/ApoEII approach produced significantly enhanced therapeutic effects, resulting in complete correction of brain pathology and behaviour.

Gene editing techniques such as CRISPR-Cas have also been introduced to HSC-GT as an alternative to traditional viral vector-based gene augmentation. With this approach, the addition of the therapeutic gene is achieved by delivering the Cas9 nuclease and a short guide RNA (sgRNA) targeting a nonessential genomic sequence (a “safe harbour”), where an expression cassette (the therapeutic gene with a promoter) will be inserted. In vitro studies have already provided evidence for successful genetic modification of HSCs using CRISPR-Cas9 [108,109]. Very recently, two studies reported pre-clinical utilisation of CRISPR-Cas9 HSC-GT for MPS I [110] and Gaucher disease [111], which will be further discussed in the Gene Editing section of this review. These two studies have established a novel approach for HSC-GT, which is highly flexible and could serve as a platform to obtain supraphysiological expression of a therapeutic protein. Moreover, compared with HSC-GT driven by RV/LV vectors, it reduces the potential risk of toxic random genome integration, and the amount of transgene expression is more predictable and consistent because of limited insertion sites. However, off-target editing remains a concern.

In summary, HSC-GT is a valuable gene therapy approach for LSDs. Gene therapy overcomes limitations of allogeneic transplantation and allows amplified expression of the therapeutic gene. Cross-correction via the M6P-R-pathway which is used by many lysosomal proteins, makes it possible to treat disease by correcting a proportion of deficient cells, especially when the therapeutic protein is overexpressed in those cells via gene therapy. In the CNS, HSC-GT is a potentially good option compared to ERT since the recombinant enzyme does not cross the BBB, and to conventional HSCT which shows limited efficacy in the CNS. HSC-GT research is particularly important for LSDs involving extensive pathologies in both the CNS and the periphery, whereas for disease predominantly affecting the brain rather than other organs, direct gene therapy targeting the brain might be more beneficial. Several attempts have been made to improve HSC-GT, as discussed above, and promising results have emerged and paved the way. In the future, several uncertainties or limitations will need to be better addressed: long-term efficacy follow-up and safety analysis, particularly with regard to viral genome integration and transgene overexpression, optimisation of current approaches and technology to better regulate the transgene expression and improve therapeutic effects and reducing complications related to pre-conditioning for transplantation.

## 3. In Vivo Gene Therapy

In vivo gene transfer therapies are based on the ability to introduce the therapeutic genetic material into affected tissues by direct administration into the patient (Figure 2). While ex vivo gene therapy represents a feasible therapeutic approach mostly for secreted enzyme deficiencies, direct administration of gene therapy vectors has now been successfully applied in many pre-clinical and clinical studies. Notably, with significant advances in vector technology and production, in vivo gene transfer has emerged as a safe and efficient technique to target the nervous system.

### 3.1. Systemic Administration

While enzyme replacement therapy is an effective treatment option, systemic gene delivery offers the opportunity for a sustained therapeutic effect comparable to long-term effects of ERT, with a single or limited number of administrations and the additional prospect of treating the CNS. The earliest vectors used for systemic delivery were based on or derived from the retroviral Moloney murine leukaemia virus. These vectors have been used to deliver the *IDUA* gene to MPS I mice [112,113] via injection into the temporal vein of neonates. Although the visceral pathology was partially rescued, a high dose was required to achieve sufficient expression, particularly in the brain. The ability of retroviral vectors to transduce only dividing cells limits their possible application for the treatment of the CNS. Similar effects were achieved when neonatal MPS VII mice were administered a high dose vector expressing the *GUSB* gene [114,115]. Similarly, neonatal MPS VII dogs injected systemically with a retroviral vector expressing the canine β-glucuronidase gene showed significant improvement in body weight, bone and joint abnormalities, motor coordination and visual impairment [116]. While neonatal gene therapy can lead to improvements in the viscera and brain, adult systemic administration of retroviral vectors can cause extensive cytotoxic T lymphocyte immune response if not combined with immunosuppressant treatment [117].

Gammaretroviral vectors have been succeeded by lentiviral vectors based on immunodeficiency viruses and have the ability to also transduce non-dividing cells. Lentiviruses show the ability to efficiently transduce hepatocytes following systemic administration. In fact, gene therapy targeted to the growing liver exploits clonal expansion of the transduced hepatocytes, resulting in high gene expression in the liver. A lentiviral vector based on the feline immunodeficiency virus FIV was first used to deliver the β-glucuronidase gene to MPS VII adult mice with the aim of providing sustained therapeutic enzyme following liver transduction [118]. Further optimisation led to the development of lentiviral vectors based on the human HIV-1 virus and their application for in vivo pre-clinical studies. A Pompe disease mouse model was treated intravenously on the day of birth with an HIV-1 based lentiviral vector expressing human *GAA* [119]. Then, 24 weeks after administration, glycogen accumulation was partially cleared in cardiac and skeletal muscles. Importantly, neonatal gene transfer elicited only a minimal antibody response despite expressing the GAA enzyme at supraphysiological concentrations. Systemic administration of lentiviral vectors expressing the IDUA gene to both neonatal [120] and adult [121] MPS I knock-out mice resulted in decreased GAG storage and improved survival. In treated animals, 1% of physiological enzymatic activity was sufficient to normalise substrate concentrations in the urine, liver and spleen. However, six months after adult injections, enzyme-specific antibodies were found to be present, causing loss of therapeutic protein expression in evaluated tissues. Conversely, seven months post-administration, long-term expression was detected in the visceral organs of adult MPS IIIB [122] and IIIA mice injected with a lentiviral vector expressing the murine sulphamidase gene [123,124]. Interestingly, these two studies from the same research group reported contrasting results regarding the effects of the therapy on the central nervous system pathology. Partial correction of the brain pathology following systemic gene therapy was achieved when the murine β-glucuronidase gene was delivered to adult MPS VII mice up to seven months after gene transfer [125]. The authors suggested that the enzyme is mainly produced in the viscera and transferred across the BBB via the mannose-6-phosphate/insulin-like growth factor 2 receptor. In addition, the expression of enzymes from transduced cells of haematopoietic origin might have further contributed to improve CNS pathology.

Adenoviral vectors have also been employed to deliver the β-glucuronidase gene to MPS VII mice [126,127], with resulting elevation of enzymatic activity in the liver and spleen. However, systemic administration of the vector failed to improve pathology within the brain. Preferential liver transduction following IV injections of the adenoviral vector was also reported in Tay-Sachs adult mice co-administered with vectors coding for both α- and β-subunits [128]. Similarly, hepatomegaly was rescued by adenoviral vector-mediated delivery of the human *LIPA* gene in a mouse model of lysosomal acid lipase (LAL) deficiency [129]. A single intravenous injection of an adenoviral vector allowed for *GAA* expression in multiple muscle groups in a quail model of Pompe disease [130] and *GAA*-knock-out mice [131,132]. In these studies, the efficient hepatic transduction following systemic gene delivery led to secretion of the enzyme at high concentrations into the plasma of treated animals, with uptake into both skeletal and cardiac muscle.

With their wide cellular tropism, low toxicity and sustained gene-expression, recombinant AAV vectors represent a compelling candidate for systemic (IV) gene delivery. Although capsids such as AAV8 and AAV9 preferentially transduce liver and muscle [133], a more modest increase in enzyme concentrations in other tissues can also contribute to reducing the amount of accumulating substrates and treating the visceral component of the diseases in different models (Table 3). Efficient and widespread visceral expression following systemic administration is achieved in a dose-dependent manner. Recent advances in the ability to produce recombinant AAV vectors at a fairly large scale has resulted in the translation of many pre-clinical studies to clinical trials. The use of AAV systemic gene therapy in paediatric patients is possibly the first step in applying this approach for LSDs within the clinic. Sio Gene Therapies in collaboration with the National Human Genome Research Institute is investigating the efficacy of intravenous high-dose (1.5 × 10^13^ to 4.5 × 10^13^ vg/kg) AAV9 gene therapy in infantile (6–12 months) type I and young (2–12 years) type II GM1 gangliosidosis patients (NCT03952637). Despite the challenges associated with manufacturing of the ever larger batches of the viral vector required for intravenous delivery in the clinic, several clinical trials involving adult patients are currently on-going. Adult Fabry disease patients have received AAV6 (NCT04046224) or other engineered capsids (NCT04040049, NCT04519749) with the aim to test safety and tolerability of the treatment following a single intravenous injection. Similarly, two phase I/II clinical trials delivering AAV8 to adults with late onset Pompe disease (NCT04093349, NCT04174105), and a self-complementary AAV9 vector to MPSIIA patients (NCT04088734 and long-term follow up NCT02716246) are now active and remain ongoing.

The ability of AAV vectors, particularly AAV9, to transduce neurons following systemic administration to animal models [157] and the recent commercialisation of Zolgensma for spinal muscular atrophy in paediatric patients have provided significant evidence to justify the use of AAV vectors as a therapeutic option for LSDs with neuropathology. However, the dose required to achieve sufficient and extensive CNS expression via intravenous administration is high and raises concerns around toxicity [158]. Severe adverse events have been reported in different clinical trials where relatively high doses of the vector (6.7 × 10^13^–3 × 10^14^ vg/kg) were administered to neonates and children, ranging from elevated serum transaminase in the SMA1 Novartis trial (NCT03306277), complement activation and acute kidney injury in the Duchenne Muscular Dystrophy trial (NCT03362502) to death from sepsis [159] in the recent ASPIRO trial (NCT03199469). As a result of a T cell-mediated immune response to high amounts of capsid antigen, hepatotoxicity is the most common adverse effect in systemic gene therapy trials [160]. This immune response is not always evident in animal models; therefore, pre-clinical studies might fail to predict the adverse effects developed by patients during trials.

Although systemic administration is arguably the most feasible approach to target diseases caused by deficiencies in secreted enzymes, gene delivery has also been used to successfully treat lysosomal defects involving membrane-bound proteins. Neonatal NPC mice received a retro-orbital injection of AAV9 vectors expressing the human NPC1 gene [161]. Treatment with vectors containing both the neuronal-specific and the ubiquitous promoter vector resulted in increased life span, reduction in cholesterol storage in the liver and a variable degree of cholesterol reduction in the brain.

### 3.2. Organs as Enzyme Factories for Systemic Expression

#### 3.2.1. Liver

The selective targeting of organs in order to produce and secrete therapeutic proteins into the bloodstream is an attractive approach to treat systemic diseases such as LSDs. Highly vascularised organs, such as the liver or the lungs, are prime candidates for this gene transfer modality. The development of a successful gene therapy for haemophilia A [62] and B [162] demonstrated that hepatocytes can be easily transduced following intravenous injection, ensuring safe and long-term transgene expression. Systemic delivery has proved to be a non-invasive delivery route, which provides the same liver transduction efficiency as direct intrahepatic administration [163]. In addition, the intravenous route allows the delivery of high doses of vectors, therefore achieving widespread and efficient transduction. However, this can come at the cost of a possible immune reaction [164,165].

While the initial choice of serotype was the early characterised AAV2, other capsids such as AAV8, AAV3B and novel engineered vectors including LK03 [166] or NP59 [167] have recently shown superior transduction efficiency in mouse and human hepatocytes. This approach was successful in rescuing a mouse model of Pompe disease following systemic delivery of an AAV8 vector that expressed an engineered acid-α-glucosidase protein secreted by hepatocytes [168]. Freeline Therapeutics has recently used both AAV8 and novel capsid AAVS3 to deliver a *GBA* vector to cells and mice in a proof-of-concept study, supporting the development of a liver-directed therapy for Gaucher disease [169]. In addition, liver-specific promoters have been designed to further improve efficacy and safety, as demonstrated in pre-clinical studies in Fabry disease [170,171], Gaucher disease [172] and Pompe disease [173,174] mouse models. Notably, systemic administration of low-dose AAV8 containing a liver-specific promoter did not result in a detectable immune response in Pompe mice, in contrast with the elevated rate of anti-GAA antibody formation following ERT administration [175]. The safety and bioactivity of an AAV8 vector expressing the human *GAA* gene under control of a liver-specific promoter are currently being assessed in a clinical trial on late-onset Pompe patients (NCT03533673). A similar approach has been tested in MPS I cats [176], and MPS VI mice [177] and cats [178] where gene expression was controlled by the thyroxine binding globulin (TBG) liver promoter. Gene therapy reduced GAG accumulation and corrected the aortic valve lesions in treated cats. The authors briefly addressed the possible effects of AAV8-mediated liver-directed gene therapy in the CNS of MPS I cats. Interestingly, although GAG storage was substantially improved in the meninges, there was no detectable improvement in the brain parenchyma and corneas of injected animals. This AAV8-based vector has been administered to MPS IV patients in a multi-centre phase I/II clinical trial (NCT03173521).

Intraparenchymal administration of an AAV2 vector expressing the murine β-glucuronidase gene to MPS VII mice resulted in increased quantities of enzyme in the viscera as a consequence of both direct transduction of extrahepatic organs and circulation of the enzyme produced in the liver through the bloodstream [179]. Interestingly, a high dose of vector partially improved the pathology in some brain regions, reducing storage accumulation in the cortex, corpus callosum and striatum.

For devastating diseases with rapid progression such as most acute forms of LSDs, early intervention is essential to prevent the development of severe pathology and ultimately early death. As already mentioned, viral gene delivery to paediatric patients presents the advantage of preventing or slowing the accumulation of macromolecules at an early stage, when the pathology is not extensive and the progression might still be reversible. Additionally, neonates are more likely to have not yet developed antibodies [180] or T cell immunity to the viruses, allowing safe and efficient administration of high doses of the vector. Liver-directed gene delivery to neonatal MPS I mice using retroviral vectors has been attempted in the past [113]. The early intervention led to high *IDUA* expression in the plasma, completely correcting cardiac, bone, eye and ear manifestations eight months post-administration. Rucker and colleagues raised the idea of early intervention by delivering an AAV vector expressing the human *GAA* gene to the liver of fetal Pompe mice [181].

Although prenatal and neonatal/paediatric application is an attractive prospect for liver-targeted gene therapy, one has to consider that the fast cell proliferation rate and turnover of hepatocytes can lead to a loss of vector genomes when using vectors considered to be non-integrating such as AAVs [182]. There are also safety concerns around the possible genotoxic effects of AAV-mediated gene therapy and its potential connection with hepatocellular carcinoma [183,184].

#### 3.2.2. Lungs

The lung can also be used as a metabolic factory for targeting systemic diseases, exploiting its large surface and extensive vascularisation. In fact, nasal delivery is a non-invasive technique that results in systemic gene expression and might offer the possibility of safe re-administration [185]. An adenoviral vector expressing α-galactosidase A was delivered via inhalation to a Fabry mouse model, resulting in increased enzymatic concentrations in some visceral organs, including the heart, liver and spleen [186]. Interestingly, no viral DNA was detected in the viscera, confirming the enzyme was produced and secreted from the lung and then taken up by other organs.

#### 3.2.3. Muscle

Skeletal muscle has also been considered a potential organ factory to produce lysosomal enzymes, especially as certain AAV serotypes are particularly efficient in transducing myofibers [187]. Intramuscular injections of AAV vectors have been used to deliver recombinant enzyme to the systemic circulation in murine models of MPS VII [188] and Pompe disease [189]. Overall, gene transfer resulted in strong expression in the muscle and increased amounts of enzyme in some visceral organs. However, the therapeutic efficacy was not always sufficient for reversing the pathology in organs distant from the site of injection. On the other hand, muscle-directed gene transfer resulted in long-term expression of circulating enzyme in a Fabry mouse model [190], proving that intramuscular administration might be effective for certain lysosomal disorders.

### 3.3. CNS-Directed Gene Therapy

While different gene therapy strategies have been effective in improving the majority of visceral manifestations, the severe neuropathology that characterises many LSDs remains a significant issue for the treatment of these patients. Drug delivery to the central nervous system, particularly to the brain, is challenging because of its unique complexity and limited access through physical barriers such as the skull and the BBB. In the last two decades, AAVs have been the vectors of choice for most of the CNS-directed gene therapy studies (Table 4).

The first approaches were mainly based on the direct infusion of the product into the brain parenchyma. Intraparenchymal (IC) injections have the advantage of providing targeted administration to the affected area and can overcome the challenge of the vector permeating efficiently through the BBB. On the other hand, the limited spread of the vector might not be efficient in treating disorders where the neuropathology is extensive and high concentrations of therapeutic protein expression are required in multiple brain areas. However, this drawback could be overcome by administering the therapy to multiple injection sites in the brain parenchyma, ensuring widespread expression in different regions. This method has been applied in clinical trials for MPS IIIA (NCT01474343), MPS IIIB (ISRCTN19853672), early onset MLD (NCT01801709) and late infantile NCL (NCT01161576, NCT01414985), where the intervention consisted of six or 12 vector deposits to different brain regions during a single neurosurgical session.

Delivery to the cerebrospinal fluid is an alternative approach to achieve widespread gene expression in the CNS. Intrathecal (IT) administration into the lumbar cistern has proven particularly efficient when using AAV9-based vectors [244]. While gene expression following IT injection in mice is restricted to the lower segments of the spinal cord, delivery to larger animals such as pigs [245] and NHPs [246] results in more diffuse transduction of the spinal cord and brain. However, studies conducted by Hinderer and colleagues showed that intrathecal delivery resulted in lower gene expression in the cervical section of the spinal cord and in the brain compared to other CSF delivery methods [247,248], even when the animals were placed in the Trendelenburg position after injection [249]. Nevertheless, the lumbar puncture procedure is widely used in the clinic. Due to its feasibility and reduced invasiveness, IT delivery was chosen as the route of administration in a phase I/II safety study where an AAV9-based vector expressing the *CLN6* gene was administered via lumbar puncture to patients with CLN6 type late infantile NCL (NCT02725580), following positive results in NHPs experimentation [250].

An extensive study conducted by Ohno et al. compared the kinetics of vector clearance from the CSF following infusion of AAV vectors to the lumbar cistern and the cerebromedullary cistern, or cisterna magna (CM) in NHPs [251]. This and other studies demonstrated that administration via suboccipital puncture to the cisterna magna is remarkably efficient in widely transducing the brain and spinal cord in NHPs [252,253]. In addition, the pattern of transduction in the brain of NHPs infused with an AAV9 vector via CM was identical to the one produced by systemic administration to the carotid artery, with particularly high gene expression in the brain [254]. CM and IV routes of administration have also been compared in an MPS VII dog model [227]. The CNS-directed treatment resulted in higher expression levels of GUSB following injection of both AAV9 and AAVrh10 vectors. MPS I dogs were also injected with an AAV9 vector into the cisterna magna, one month after preventive immune tolerization via systemic administration during the perinatal period [225]. The tolerised animals did not develop antibodies against the *IDUA* enzyme in the brain, allowing a safe and effective re-administration to the CSF. Together, these studies suggest that CM gene delivery can be a feasible approach for treating the neuropathology in many LSDs, with the advantage of a lower vector dose requirement and reduced exposure of the vector to the peripheral organs compared to systemic administration. In 2020, Prevail Therapeutics started recruiting type II Gaucher patients to enrol in the PROVIDE clinical study (NCT04411654), where infants diagnosed with the acute form of nGD will receive a CM injection of an AAV9 vector. However, it has been shown that CM delivery is not easily translatable to the clinic, as this technique might be associated with risks related to high vascularisation of the tissue in human patients [253,255]. In addition, there are reports of dorsal root ganglia (DRG) sensory neuron degeneration and secondary axonopathy following CM administration of AAV9 delivering human iduronidase to NHPs [252]. The possible mechanisms responsible for this reaction currently remain undefined. Hordeaux et al. down-regulated transgene expression in the DRG neurons of NHP by incorporating into the vector genome sequence targets for the microRNA miR183, which is present almost exclusively in the DRG, into the vector genome [256]. This modification resulted in reduced toxicity in the DRG, while gene expression was not affected in the rest of the CNS. Recently, Taghian and colleagues developed a novel delivery technique using a microcatheter inserted in the lumbar region that reaches the cisterna magna in the suboccipital space through the spinal canal [257]. The safety and biodistribution of this delivery approach were first evaluated in sheep. Extensive transduction of brain cortical regions and the spinal cord and modest biodistribution in the peripheral organs were detected in the animals three weeks following administration. Two infant Tay-Sachs patients were also dosed using a spinal canal microcatheter, with no reported adverse effects during the infusion or post-treatment. In 2019, Regenxbio Inc. commenced a phase I/II clinical trial (NCT03566043), where three infant MPS II patients received an AAV9 vector via CM infusion. The first results showed that the treatment was well tolerated and caused a sustained reduction of heparan sulphate in the CNS [258].

Extensive transduction of the CNS is also achieved via intracerebroventricular (ICV) administration. ICV delivery to NHPs results in larger cortical distribution of the vector compared to the intrathecal route, with substantial transduction of the cerebellum [251] and a comparable distribution throughout the brain and spinal cord to CM administration [249]. ICV administration is a relatively common technique compared to CM and it has proven particularly effective in neonatal animals, where it mediates extensive vector distribution in the parenchyma with widespread neuronal transduction and partial transduction of some peripheral organs in animal models of severe neuro-metabolic diseases [146,239]. In addition, delivery to the lateral ventricles can be efficiently combined with systemic AAV administration, resulting in further improvement of the treatment as demonstrated in a pre-clinical study developing gene therapy for multiple sulfatase deficiency [152].

A self-inactivating HIV-based lentivirus vector expressing the human arylsulfatase A *ARSA* gene was used in a pre-clinical study in metachromatic leukodystrophy [259]: 10-month-old knock-out mice received unilateral injection into the right fimbria in the hippocampus under stereotactic guidance and were examined one, three and five months later. Extensive transduction of neurons and astrocytes was detected in different areas of the hippocampus, with no major adverse effects. The treatment resulted in reduced neuronal loss and lipid deposition, increased enzyme expression and activity in both the treated and the contralateral uninjected hemisphere; and rescued short-term and long-term spatial memory. Although lentivirus-mediated transduction spread is usually limited to the focal administration area, the authors speculated that the widespread enzymatic activity was caused by the mechanism of cross-correction. Therefore, lentiviral vector-based gene therapy could be an effective therapeutic option for MLD, as less than 5% of wild-type enzymatic activity would be required to prevent the neurological symptoms [260]. This approach has been recently translated to the clinic, where a phase I/II clinical trial (NCT03725670) is currently exploring the safety and efficacy of brain-direct gene therapy using a self-inactivating lentiviral vector in MLD patients.

The characteristic temporal and spatial expression patterns can limit the applicability of lentiviral vectors for those disorders where widespread transgene expression might not be required or might cause brain toxicity. Widespread gene delivery to the brain has been achieved using a recombinant adenoviral vector expressing the human β-glucuronidase gene to the striatum of MPS VII mice [261]. Increased enzymatic activity was detected not only in many regions of the ipsilateral hemisphere but also in the liver of treated mice up to 16 weeks post-injection when administered in combination with a systemic injection [126]. However, because of the acute immune response associated with adenoviral vectors, the animals were transiently treated with the immunosuppressant MR-1. To avoid loss of transgene expression due to the immune response, researchers developed a new generation of helper-dependent vectors based on the canine adenovirus type 2 (CAV-2) [262]. These vectors do not induce adaptive cell-mediated immune response and therefore mediate long-term gene expression with limited need for immunosuppression. A CAV-2 vector was used to deliver the β-glucuronidase gene to the brain of MPS VII mice via bilateral striatal injections [263]; 16 weeks post-administration, enzymatic activity was detected in several areas of the forebrain and midbrain, with significant correction of pathology in neurons and glial cells. Preferential neuronal transduction and correction of brain pathology were also achieved in the MPS VII dog model, following brain-direct administration of the same vector [264].

A similar approach was developed for the treatment of MPS IIIA mice, where 6–18-week-old adult mice received bilateral injections into the thalamus with a CAV-2 vector expressing the *SGSH* gene [265]. However, transgene expression was dose-dependent and enzymatic activity was already undetectable two weeks post-injection. On the contrary, long-lasting expression was achieved following neonatal administration via bilateral injections into the lateral ventricles. Enzymatic activity was elevated 20 weeks post-treatment with a significant reduction in lysosomal storage and a lack of neutralising antibody formation.

Despite CAV-2 vectors having proven effectiveness in transducing neurons following brain-directed administration and their ability to promote widespread gene expression following retrograde axonal transportation [266], their application has currently not been translated to the clinic. The ability of CAV-2 to transduce neuronal cells depends on the expression of the CAR receptor, which is restricted to neurons in the mouse brain [267]. However, CAR expression varies considerably in different species and neuronal types [266]; therefore, CAV-2 vectors might not be able to efficiently transduce CAR-negative neurons in the human brain.

### 3.4. Other Organ-Targeted Gene Therapy Approaches

#### 3.4.1. Eye

While brain-directed gene therapy showed promising results for the treatment of the neurological manifestations in many LSDs, correction of the disease is usually limited to the brain. However, other structures of the CNS might be affected by substrate accumulation and cell loss. Batten disease, for example, is characterised by severe and progressive degeneration of the retina [268]. As of today, correction of retinal atrophy and consequent visual loss have not been achieved following brain-directed gene therapy. However, focal intravitreal administration of an AAV2/7m8 vector prevented loss of photoreceptor cells in the retina of a CLN6 mouse model [269]. Interestingly, direct transduction of photoreceptors did not result in any therapeutic effect, while overexpression of the CLN6 gene in the bipolar cell layer significantly slowed retinal degeneration and loss of photoreceptor functionality. This study showed that bipolar cells could be a therapeutic target for several Batten disorders, although transduction efficiency of the AAV7m8 vector in human bipolar cells has not yet been assessed. Sub-retinal injections of another novel AAV capsid (AAV-TT) resulted in higher transduction of photoreceptor cells compared to standard AAV2 [205]. A similar approach was previously adopted in pre-clinical studies with the aim of treating the progressive retinal degeneration and photoreceptor cell loss characteristic of MPS VII, using AAV-mediated intravitreal gene delivery [270,271].

#### 3.4.2. Muscle

Localised administration to the muscle has been widely used to treat skeletal myopathy caused by glycogen accumulation in Pompe disease. Several studies have shown that injection of AAV2 [189], AAV6 [272] and AAV9 [273] vectors to the tibialis anterior or the gastrocnemius muscle of *Gaa* knock-out mice led to glycogen clearance and amelioration of the neuromuscular phenotype. However, prolonged overexpression of *GAA* is associated with transgene immunogenicity. The immune reaction was successfully attenuated using immune deficient *Gaa*-KO/SCID mice [272]. A similar approach was adopted in a current clinical trial (NCT02240407), assessing the safety of two consecutive administrations of an AAV9 vector to both legs of Pompe patients under a regime of immunosuppressive drugs.

Pompe disease is characterised by weakening of the diaphragm and other respiratory skeletal muscles, resulting in ventilatory insufficiency [274]. Although direct gene delivery was efficient in increasing GAA enzymatic activity in the leg muscle, intramuscular injections did not produce any effect in the diaphragm of treated animals. An interesting gene delivery method that combines an AAV2 vector and glycerin-based gel has been successfully used to transduce the diaphragm of *Gaa* knock-out mice [275]. Topical delivery of the vector resulted in substantially increased enzymatic activity in the diaphragm of treated animals and improved ventilatory function. However, as with the intramuscular injections, early intervention in young mice was more efficient in normalising the pathology compared to the treatment of older animals. The safety of intramuscular administration to the diaphragm of Pompe patients was also investigated in a phase I/II clinical study (NCT00976352) using an AAV1 vector to deliver the therapeutic gene to late infantile/juvenile subjects.

In addition, Pompe patients often present with tongue weakness and typically develop pharyngeal dysphagia due to hypoglossal XII motor neuron dysfunction [276]. In an attempt to target the TXII motor neurons, ElMallah and colleagues delivered AAV1 and AAV9 vectors to the tongue of *Gaa* knock-out mice [277]. Exploiting the retrograde axonal transport to the neurons, gene therapy resulted in overexpression of *GAA* and the clearance of glycogen storage in XII motor neurons, with consequent improvement in the upper airway motor function of treated mice.

Further research supported the contribution of the CNS to the respiratory muscle impairment in Pompe disease [278]. With the aim of targeting the respiratory and cardiac dysfunctions, while potentially delivering the functional enzyme to the phrenic and intercostal motor neurons via retrograde transportation, the authors administered an AAV9 vector intrapleurally to adult *Gaa* knock-out mice. Six months post-injection, enzymatic activity was significantly improved in the lungs and myocardium of treated mice, with consequent clearance of glycogen storage. Interestingly, vector genome copies were also detected in the diaphragm and spinal cord, supporting the hypothesis that transduction of respiratory motor neurons can be beneficial in improving respiratory function in Pompe disease.

## 4. Gene Editing

The gene therapy approaches mentioned so far in this review rely on gene augmentation, rather than correction of the detrimental mutations in the patients’ genome. Genome editing aims to manipulate the patient’s genome in a controlled manner. For example, it may allow insertion of the desired gene at a specific site within the host genome. When targeting actively dividing cells, this would overcome the problems of dilution of a non-integrating vector and the potential genotoxic effects of uncontrolled insertion [160]. The ultimate goal of gene editing is to be able to selectively correct a harmful mutation in the patient’s own genome to the correct nucleotide base sequences.

### 4.1. Double-Strand Breaks (DSBs)

Gene editing involves the manipulation of the cell system for repairing double-strand breaks in the genome. Two of the most common methods are non-homologous end joining (NHEJ) and homology directed repair (HDR). The NHEJ repair mechanism mediates the direct re-ligation of the broken DNA without the need of a homologous template to guide repair. The NHEJ mechanism is the quickest at restoring the DNA strand; however, the repair mechanism is highly error prone and sequence independent, frequently leading to small insertions and deletions (indels) [279]. This can be beneficial when trying to create a knock-out model, as it can lead to a reading frame shift and the formation of an early stop codon. However, NHEJ is unsuitable in the therapeutic setting, as the random nature of the system means that the end population of cells could exhibit a diverse array of mutations [280,281,282].

HDR is a naturally occurring system which can repair DNA DSBs with high efficiency. HDR requires a homologous DNA repair template, which contains the desired edit within stretches of the homologous sequence both immediately upstream and downstream of the target site. These are typically ~1.2 kb in length for mammalian cells [279,283,284]. A Holliday junction is created when there is sufficient homology for the template strand to displace the host strand and form a single strand cross over [285]. This is then cleaved by endonuclease GEN1 and the strands are fused together by a ligase. This method of repair only occurs during the S and G2 phases of the cell cycle, thus limiting this mechanism to actively replicating cells [286].

### 4.2. Zinc Finger Nucleases (ZFNs)

Zinc finger nucleases (ZFNs) are synthetic, engineered proteins consisting of the non-specific cleavage domain of FokI endonuclease and zinc finger proteins [287]. These motifs are able to bind to DNA by inserting into the major groove of the double helix and recognising specific base triplets [288]. When ZFNs bind to unique sequences within the genome, the FokI nucleases create a double-strand break at the target site allowing for local homologous recombination and consequent targeted delivery of a therapeutic gene [289].

In the LSD field, approaches with ZFNs have been explored for Gaucher disease [290], Fabry disease [290], MPS I [290,291] and MPS II [290,292]. Wild-type mice were treated with 3 × 10^11^ vg of both AAV8-ZFNs and 1.5 × 10^12^ vg AAV8-donors which contained either acid-β-glucosidase (Gaucher), α-galactosidase A (Fabry), α-L-iduronidase (MPS I) or iduronate-2 sulfatase (MPS II). The ZFNs were designed to induce a double-strand break in the albumin gene, while the donor genes were flanked by homologous repeats and a splice acceptor sequence. Four weeks post-administration, supraphysiological concentrations of enzymes were detected in the liver, while a 3-fold increase in enzymatic activity persisted for eight weeks post injection. Sequencing analysis of the genomic DNA showed a 31.4% modification at the intended target with less than 2% at queried off target loci.

α-L-iduronidase *IDUA* knockout mouse models, aged 4 to 10 weeks old, were intravenously injected with 1.5 × 10^11^ vg of each AAV8-ZFN targeted at intron 1 of the albumin locus and 1.2 × 10^12^ vg AAV8-IDUA donor [291]. This approach led to 34–47% insertions by one month post-injection. The *IDUA* activity in the liver was 10- to 16-fold higher compared to wild-type controls, and enzyme activity in the plasma was 7- to 9-fold higher compared to the control mice at 28 days post injection. Secondary tissues including the spleen, heart, lungs and muscle also showed an increase in *IDUA* activity one month post-administration; however, no significant increase was observed in the brain. This suggested that uptake by the M6P-R pathway allowed for a degree of cross correction to occur, as confirmed by the reduction of the glycosaminoglycan concentrations in the liver and all peripheral tissues at the four-month time point.

Similar experiments were also performed on a mouse model of MPS II [292]. Male mice between six and nine weeks old were intravenously injected with AAV8 ZFN vectors and the AAV8 IDS donor at three doses, with the highest being 1.5 × 10^11^ vg AAV-ZFN and 1.2 × 10^12^ vg AAV-donor. The enzyme concentration in the liver was 207-fold higher compared to the wild-type mice at four months post-injection, with the GAG concentration restored to wild-type levels.

In a clinical trial for the treatment of MPS I (NCT02702115), three participants were enrolled in a phase 1/2 trial to receive ascending single doses of SB-318 (a three-component therapeutic of AAV6-ZFN1, AAV6-ZFN2 and AAV6-IDAU donor) administered by intravenous infusion. The purpose of this study was to assess the safety and tolerability of the agent, with the secondary outcomes being measured including the change in *IDUA* activity, the change of urine GAG levels and AAV6 clearance. The trial sponsored by Sangamo Therapeutics has been terminated as the study did not demonstrate a clear clinical benefit.

### 4.3. Transcription Activator-Like Effector Nucleases (TALENs)

The transcription activator-like effector (TALE) protein is derived from the pathogenic plant bacteria *Xanthomonas* [283,293]. The protein has repeats of a DNA binding module, consisting of 34 amino acids that allow specific single DNA base recognition. These DNA binding domains are able to be synthesized in tandem to form an array that can recognise a specific target sequence with more flexibility than ZFNs due to single base pair reading compared to triplets. TALE proteins are fused with FokI nucleases. This allows FokI to dimerise upon binding to the target DNA and create a double-strand break at a targeted location within the genome [294,295]. Exogenous DNA can be inserted at the DSB site via a non-homologous end-joining repair mechanism, using transfected dsDNA sequences as a template for the repair. The use of TALENs with regards to LSDs has been limited so far, having only been used to produce models of Gaucher disease in human-induced pluripotent stem cells and zebrafish [282,296].

### 4.4. Clustered Regularly Interspaced Short Palindromic Repeats (CRISPR)

A Nobel Prize winning advancement in the field of gene editing was the discovery of the CRISPR system [297]. Bacteria have an RNA-mediated adaptive immune system based on clustered regularly interspaced short palindromic repeats (CRISPR). When foreign pathogenic DNA is detected within the cell, short strands of the invading DNA are incorporated into the bacterial genome between CRISPR repeat sequences [298]. The CRISPR-associated protein 9 (Cas9) is an endonuclease that recognises CRISPR repeats and cleaves specific complementary DNA sequences. By complexing Cas9 with a synthetic guide RNA sequence (gRNA), which targets a specific genomic sequence and forms a scaffold for the Cas9 enzyme, the CRISPR technology can be used to modify, remove or add genes in vivo.

The use of CRISPR in the field of LSDs has been explored for the creation of novel disease models [282] and the development of potential therapeutic strategies, where a therapeutic cDNA sequence is inserted at different loci within the genome, such as the albumin locus, acting as a safe harbour. One of these strategies, for the treatment of MPS I, follows on from previous work using ZFNs to try and insert the *IDUA* gene into the albumin locus [291]. A system was designed using CRISPR technology to create a DSB in the albumin gene, allowing for insertion of the *IDUA* cDNA [299]. Neonatal MPS I mice were injected intravenously with two AAV-based vectors, one carrying the CRISPR/Cas9 system and the other the donor *IDUA* sequence. At 11 months post-administration, a significant increase in *IDUA* activity was observed in the liver, heart, spleen and brain tissue of the animals treated with the highest dose; 3 × 10^14^ vg/kg AAV-IDUA and 5 × 10^13^ vg/kg AAV-Cas9. These encouraging results were also reflected in the GAG concentrations with a significant decrease in the treated mice compared to the untreated controls. Examination of the tissue by light microscopy revealed a reduced incidence of foam cells in the liver and decreased vacuolation of Purkinje cells in the brain of treated mice. Noticeable behavioural improvement was observed, with treated mice showing restored memory and learning ability.

Another similar approach using the albumin locus as a safe harbour has also been investigated for the treatment of Tay-Sachs and Sandhoff diseases [300]. The inserted gene was a modified human Hex hybrid µ-subunit (*HEXM*), which incorporates the active site of subunit-α and stable subunit-β interface along with areas from each subunit required to interact with the GM2 activator protein. Together, these subunits form a homodimer capable of degrading GM2 gangliosides [210,301,302]. Neonatal Sandhoff mice were injected with a dual AAV system consisting of AAV8-SaCas9 (5 × 10^9^ vg/g) and AAV8-HEXM-sgRNA (3 × 10^10^ vg/g) via the temporal facial vein. One month post administration, the plasma 4-methylumbelliferyl-β-*N*-acetylglucosamine-6-sulphate (MUGS) and 4-methylumbelliferyl-β-*N*-acetylglucosamine (MUG) activity increased 144-fold and 17-fold, respectively, when compared to the wild-type. The tissues were harvested following euthanasia at four months of age and assessed for enzymatic activity. There was a 7-fold increase in the liver of treated mice compared to the wild-type, and a significant increase in the brain, heart and spleen compared to the untreated group. The rotarod results showed that there was also a significant locomotor improvement in the treated mice.

A frequent mutation in South American MPS I patients is a 1205G>A mutation which leads to the formation of a premature stop codon [282,303]. The use of the CRISPR/Cas9 technology was combined with the provision of a single strand of wild-type donor sequence for *IDUA* to allow for HDR. The agents were first delivered to human fibroblasts using a non-viral nano-emulsion system, and then to new-born MPS I mice via systemic administration. The serum *IDUA* activity was restored to around 6% of normal, and the enzymatic activity in the peripheral organs examined showed a significant increase compared to the untreated controls.

Ex vivo gene editing has also been evaluated for the treatment of MPS I [110,304]. In order to generate human CD34+ haematopoietic stem cells, sgRNA for the *CCR5* gene and Cas9 protein were electroporated into HPSCs followed by AAV6-mediated transduction for delivery of the homologous templates [110]. The *CCR5* site was chosen as the safe harbour, because it is a non-essential gene which poses no detrimental effect at deficiency. The frequency of modification was reported as 54 ± 10% for cord blood-derived HSPCs, and 44 ± 7% for peripheral blood-derived HSPCs. The *IDUA* gene was placed under the control of the spleen focus forming virus (SFFV) or phosphoglycerate kinase promoters, which led to a 250-fold and 50-fold increase in enzymatic activity, respectively. Upon co-culture with MPS I patient-derived fibroblasts, the modified HSPCs led to a decrease in lysosomal compartment size by means of cross correction. Subsequently, homozygous NOD-scid-gamma (NSD)-IDUA^X/X^ mice were created by CRISPR-Cas9 as a model of MPS I. At 18 weeks post-engraftment, the GAG concentrations in urine had reduced by 65% compared to the sham-treated mice.

The research group then established this approach for potential treatment of another LSD; Gaucher disease [111]. Again, the *CCR5* gene was selected as the safe harbour, and the glucocerebrosidase expression cassette was inserted at the targeted site. The modified HSCs were confirmed for their capacity for long-term engraftment and multi-lineage differentiation in the injected mice, as well as confined glucocerebrosidase overexpression specific to the monocyte/macrophage lineage.

### 4.5. Base Editing

A further iteration of the CRISPR/Cas9 system has been the development of base editors, which come in two classes: cytosine base editors (CBEs) [305], which are capable of converting C•G base pairs to T•A, and adenine base editors (ABEs) [306], which convert the base pair A•T to C•G. These proteins have been engineered to perform transition mutations without the formation of a DSB [279].

One of the factors which has limited the use of base editors for in vivo work so far, has been the size of the DNA strand encoding the editor [307]. At 5.2 kb for the editor alone without any of the regulatory elements or gRNA, this exceeds the carrying capacity of AAV capsids. A method to overcome this limitation has been developed where the editor is split in half and fused to a fast-splicing split intein. Inteins are segments of a protein sequence located at a splicing site, which are removed from the precursor peptide allowing for ligation of the remaining adjacent regions [308]. The two halves can then be delivered by a dual AAV vector system. Following transduction of a single cell by both AAV vectors and expression of the complementary gene sequences, protein splicing occurs and all exogenous sequences are removed, leaving the base editor in its original form.

This system was then explored for its therapeutic potential for the treatment of Niemann-Pick type C disease [307]. The T3182C mutation was selected as the target, and P0 pups of a NPC1 mouse model were injected retro-orbitally with the dual AAV system at two distinct doses, with the highest being 5 × 10^10^ vg of each AAV. The treatment led to a 9.2% increase in lifespan compared to the untreated mice. At around 100 days post-injection, the number of surviving Purkinje neurons was found to have modestly increased in the treated mice to 38% of wild-type levels. In order to examine the extent of editing efficiency, the brain tissue was also examined, with nearly 50% of the cortical neurons been edited.

### 4.6. Prime Editing

Base editors are a powerful tool in the genome editing toolkit; however, they are limited to transition purine-to-purine or pyrimidine-to-pyrimidine mutations. Prime editing is a novel technology capable of making a wider range of changes to the genome such as targeted insertions, targeted deletions, all four transition mutations and all eight transversion mutations, either alone or in combination [309]. Prime editors consist of a reverse transcriptase fused to an RNA-programmable nickase, which functions with a prime editing guide RNA (pegRNA). The pegRNA acts in a manner similar to that of the guide RNA in the CRISPR/Cas9 system, where the sequence hybridizes to the target DNA and act as the template strand for the reverse transcriptase component of the prime editor.

With regards to LSDs, the most common mutation which causes Tay-Sachs disease is a TATC insertion into the *HEXA* gene at the 1278 position [309]. Anzalone et al. initially used the third-generation prime editor (PE3) to install the 4-bp insertion into the *HEXA* gene of HEK293T cells, with a 31% efficiency and 0.8% indel rate. The authors then isolated two cell lines that were homologous for the mutated gene and attempted to rectify the mutation. The most successful correction to wild-type *HEXA* had 33% efficiency and 0.32% indel formation [309]. These results supported the possible use of prime editing to correct pathogenic genetic variants; however, strategies to improve delivery to the target cells and an increase the editing efficiency will have to be further investigated before translating this technology to a clinical setting.

## 5. Conclusions

Lysosomal diseases have proven to be fertile ground for the development of gene therapy. This is understandable given the number of conditions, knowledge of the defective gene, the availability of animal models for pre-clinical studies and the frequent absence of effective treatments. Some of these conditions may be considered ‘lower hanging fruit’, where the gene therapy approach would benefit from secretion of the protein or enzyme from transduced cells and uptake by other untransduced cells. However, improvement in the viral vector technology and a better understanding of the appropriate route of administration now permit targeting of more difficult conditions. These involve non-secreted or membrane-bound proteins where efficacy is more dependent on the transduction efficiency of the viral vector in the relevant organs. Engineering of the vector and expression cassette can improve transduction efficiency and penetration into specific tissues. This is particularly important for LSDs with a neurological component, where efficient administration of the vector to the CNS is essential and may also require supplementation via other routes of delivery.

While gene therapy is an attractive therapeutic option, ethical concerns related to the possible risk profile require careful consideration. The absence of alternative treatments is clearly a strong argument in favour of gene therapy; however, the risk of extending the life span of the patient while only partially restoring their pathology without significant improvements in their quality of life, is still a critical concern. Therefore, scientists and clinicians should be mindful and carefully evaluate suitability of the therapy to protect both patients and caregivers’ interests.

Although LSDs are rare disorders, the large number of gene therapy companies invested in the LSD field is encouraging and also gives a sense of the optimism for future licensed therapies for conditions with unmet medical needs. In addition, progress in developing novel therapies proceeds in parallel with the assessment of new biomarkers necessary to diagnose new patients earlier and more efficiently and to monitor the course of the treatments over time. Overall, this advancement means that the outcome of the ongoing and soon to be initiated clinical trials is of vital importance, first to the patients and their families, but also for future development of novel advanced therapy medicinal products in other fields other than gene augmentation therapy for LSDs.

## Figures and Tables

**Figure 1 biomolecules-11-00611-f001:**
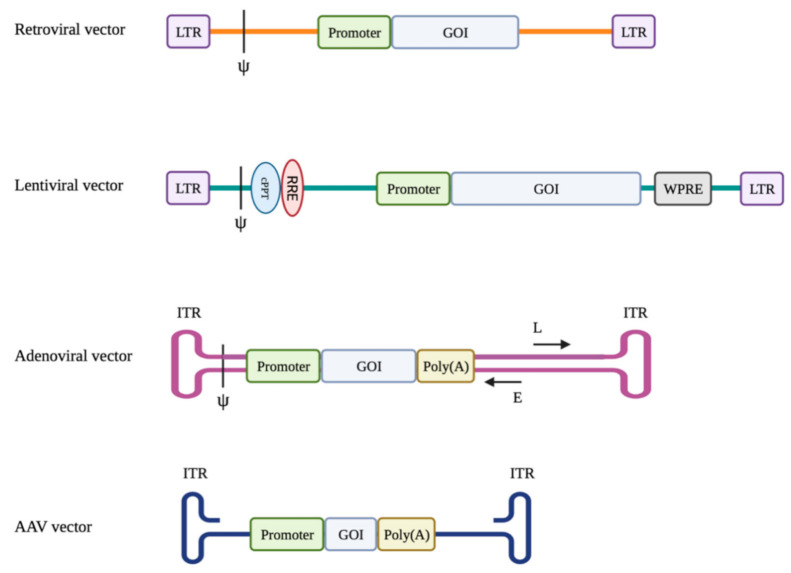
Schematic representation of the vector genomes (LTR: Long Terminal Repeat; Ψ: packaging signal; GOI: Gene of Interest; cPPT: central Polypurine Tract; RRE: rev-binding element; WPRE: Woodchuck Post-transcriptional Regulatory Element; ITR: Inverted Terminal Repeat; L: Late adenoviral genes; E: Early adenoviral genes).

**Figure 2 biomolecules-11-00611-f002:**
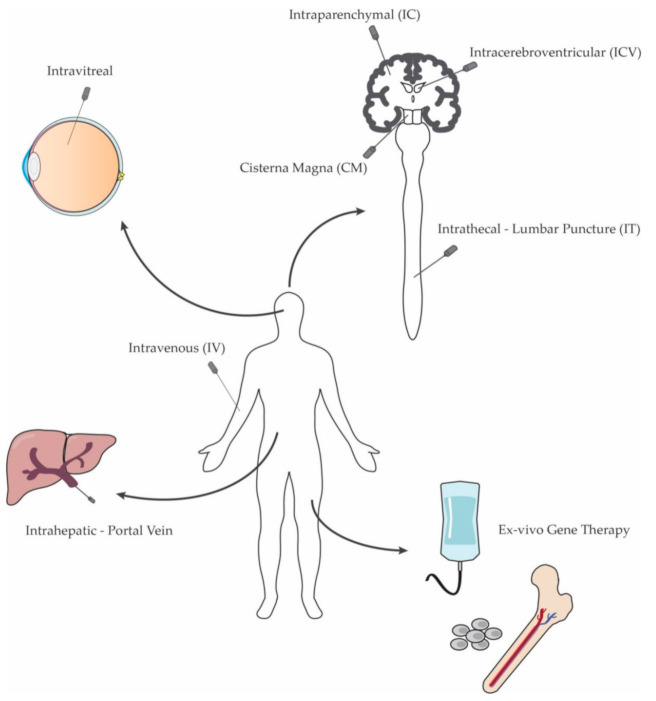
Main routes of administration for in vivo gene delivery to LSDs patients.

**Table 1 biomolecules-11-00611-t001:** Characteristics of the most used viral vectors in gene therapy for LSDs.

Vector	Size (nm)	Genome	Packaging Capacity (Kb)	Cell Type	Integration Profile
Retroviral	100	RNA	8	Dividing	Integrating
Lentiviral	100	RNA	14	Dividing and non-dividing	Integrating
Adenoviral	70–100	dsDNA	8–10	Dividing and non-dividing	Non-integrating
AAV	25–20	ssDNA	4.7	Dividing and non-dividing	Non-integrating

**Table 2 biomolecules-11-00611-t002:** HSC-GT clinical trials recruiting LSDs patients or currently active (LV: lentiviral vector).

Disease	NCT Ref.	Intervention	Status	Est. Participants
MLD	NCT01560182	OTL-200 (autologous CD34^+^ enriched cell fraction transduced with human *ARSA* LV)	Phase 1/2; Active, not recruiting	20
MLD	NCT03392987	OTL-200	Phase 2; Active, not recruiting	10
MLD	NCT04283227	OTL-200	Phase 3; Recruiting	6
MLD	NCT02559830	Transduced CD34^+^ autologous HSCs	Phase 1/2; Recruiting	50
MPS IIIA	NCT04201405	Autologous CD34^+^ cells transduced with the human *SGSH* LV	Phase 1/2; Recruiting	5
MPS I-H	NCT03488394	Autologous CD34^+^ cells transduced with human *IDUA* LV	Phase 1/2; Recruiting	8
Cystinosis	NCT03897361	CTNS-RD-04 (autologous CD34^+^ enriched cells transduced with human *CTNS* LV)	Phase 1/2; Recruiting	6
Fabry	NCT03454893	AVR-RD-01 (autologous CD34^+^ cell-enriched fraction transduced with human *AGA* LV)	Phase 1/2; Recruiting	12
Gaucher	NCT04145037	AVR-RD-02 (autologous CD34^+^ enriched HSCs transduced with human *GBA* gene LV)	Phase 1/2; Recruiting	16

**Table 3 biomolecules-11-00611-t003:** Overview of some AAV pre-clinical studies using systemic gene delivery to treat various animal models of LSDs (vg: vector genomes; gc: genome content, equivalent of vg; iu: infective units; P: Post-natal day).

Delivery Route	Disease	Species	Model	Vector	Dose	Time of injection	Ref.
Tail Vein							
	Gaucher	Mouse	UBC-creERT2-Gba^flox/flox^	AAV9-CMV-Gba or AAV9-SYN-Gba	3–5 × 10^11^ vg	4 weeks	[134]
	GM1	Mouse	C57BL/6J-*βgal*^−^^/−^	AAV9-CAG-βgal	1 × 10^11^ or 3 × 10^11^ vg	6 weeks	[135]
	GM2 (Sandhoff)	Mouse	C57BL/6:C129-*Hexb*^−/−^	AAV9-CMV-HEXB	2.5 × 10^14^ or 3.5 × 10^13^ vg/kg	P1 or 6 weeks	[136]
	GM2 (Sandhoff)	Mouse	Hexb^−/−^	AAV9 or PHP.B-BiCBA-mHEXA/mHEXB, AAV9 or PHP.B-P2-HEX	AAV9: 1 × 10^12^–4 × 10^12^ vg; PHP.B: 3 × 10^11^–1 × 10^12^ vg	4 weeks	[137]
	Krabbe	Mouse	C57BL-*twi/twi*	AAVhr10-CAG-mGALC	2 × 10^11^ vg	P10–12	[138]
	MPS I	Mouse	C57BL/6-*Idua*^−/−^	AAV9-CAG-IDUA or AAVrh10-CAG-IDUA	1 × 10^12^ vg	8–21 weeks	[139]
	MPS II	Mouse	*ids^y/-^*	AAV8-EF1α-IDS	1 × 10^11^ vg	NA	[140]
	MPS IIIA	Mouse	C57BL/6-*Sgsh*^−/−^	scAAVrh74-U1a-hSGSH	5 × 10^12^ vg/kg	4–6 weeks	[141]
	MPS IIIA	Mouse	C57BL/6-*Sgsh^G91A^*	AAV9-CAG-mSGSH	1 × 10^12^ vg	2 months	[142]
	MPS IIIB	Mouse	C57BL/6-*Naglu*^−/−^	AAV9-CAG-hNAGLU	1 × 10^13^ vg/kg	4–6 weeks	[143]
	MPS IIIB	Mouse	C57BL/6-*Naglu*^−/−^	AAV2-CMV-hNAGLU	4 × 10^11^ vg	4–6 weeks	[144]
	MPS VII	Mouse	b6.C-H-abm1/byBir-gus^mps/mps^	AAV9-βGluc	1 × 10^12^ vg	6 weeks	[145]
Temporal/Cephalic Vein							
	Gaucher	Mouse	CD1-K14-ln/ln	AAV9-GUSB-hGBA1	4 × 10^11^ gc	P0–P1	[146]
	Gaucher	Mouse	CD1-K14-ln/ln	AAV9-SYN-hGBA1	2.4 × 10^12^ vg	P0–P1	[147]
	MPS I	Mouse	*Idua* ^−/−^	AAV2-CAG-hIDUA	10^10^ vp	P1–P2	[148]
	MPS II	Mouse	*ids^y/-^*	AAV5-CMV-hIDS	1 × 10^11^ vg	P2	[149]
	MPS VII	Mouse	B6.C-H-2bm1/ByBir-gus^mps/mps^	AAV-CAG-hGUSB	5 × 10^9^ iu/kg	P2	[150]
	MPS VII	Mouse	B6.C-H-2bm1/ByBir-gus^mps/mps^	AAV-CAG-hGUSB	8 × 10^10^ iu/kg	P2	[151]
	MSD	Mouse	C57B6/S129J-*Sgsh*^−/−^	AAV9-CMV-hSUMF1	2 × 10^11^ vg	P1	[152]
Jugular Vein							
	MLD	Mouse	C57BL/6-*ASA*^−/−^	AAV9-CAG-hARSA	2 × 10^12^ vg	P1–2	[153]
Retro-orbital Sinus							
	Pompe	Mouse	C57BL/6J-GAA^−/−^	AAV-PHP.B-CAG-hGAA	5 × 10^12^ vg/kg	2 weeks	[154]
Combination							
IV + ICV	MSD	Mouse	C57B6/S129J-*Sgsh*^−/−^	AAV9-CMV-hSUMF1	IV: 2 × 10^11^ vg; ICV: 6 × 10^9^ vg	P1	[152]
IV + IT	MPS VII	Mouse	C57BL6/J-Gus^b/b^	AAV2-CMV-GUSB	IV: 4.1 × 10^11^ vg; IT: 3 × 10^10^ vg	P3–P4	[155]
IV + CM	MPS IIIB	Mouse	C57BL/6-*Naglu*^−/−^	AAV2-CMV-hNAGLU	IV: 4 × 10^11^ vg; CM: 5 × 10^10^ vg	4–8 weeks	[156]

**Table 4 biomolecules-11-00611-t004:** Selection of the most relevant pre-clinical studies on CNS-directed gene therapy in animal models of LSDs using AAV vectors (vg: vector genomes; gc: genome content, equivalent of vg; vp: viral particles; iu: infective units; pu: purified units; P: Post-natal day; E: Embryonic day; NHP: non-human primate).

Delivery Route	Disease	Species	Model	Vector	Dose	Time of injection	Ref.
Intraparenchymal (IC)							
	CLN1 Batten	Mouse	C57BL/6-*Ptt1*^−/−^	AAV9-CAG-hPTT1	1 × 10^12^ vg/mL	P1	[191]
	CLN2 Batten	Mouse	C57BL/6-*CLN2*^−/−^	AAV2 or AAV5-CAG-hCLN2	3.6 × 10^9^ gc	6 or 10 weeks	[192]
	CLN2 Batten	Mouse	C57BL/6:129Sv-*TPPI*^−/−^	AAV2 or AAV5-CAG-hCLN2	1.2–3.6 × 10^9^ gc	6 or 10 weeks	[192]
	CLN3 Batten	Mouse	C57BL/6N-*Cln3*^Δex7/8^	AAVrh10-CAG-hCLN3	3 × 10^10^ gc	P2	[193]
	Gaucher	Mouse	GBA^L444P/L444P^	AVV5-CAG-hGBA	3.5 × 10^13^ vg/mL	8 months	[194]
	GM1	Mouse	C57BL/6-*βgal*^−/−^	AAV1-CAG-mβgal	4 × 10^10^ gc	6–8 weeks	[195]
	GM2 (Sandhoff)	Mouse	C57BL/6-*Hexb*^−/−^	AAV1-HEXA and AAV1-HEXB	9.9 × 10^9^ and 1.4 × 10^10^ gc	1 month	[196]
	Krabbe	Mouse	C57BL/6-twi^W339X^ or C57BL/129SVJ/FVB/N-twi-trs	AAV1-CMV-mGALC	3 × 10^10^ vp	P0–P1	[197]
	Krabbe	Mouse	B6.CE-*Galc*^twi^/J	AAV2 or AAV5-CAG-GALC	5.5 × 10^6^ iu	P3	[198]
	MLD	Mouse	C57BL/6J-*ASA*^−/−^	AAVrh10-CAG-hARSA or AAV5-PGK-hARSA	2.3 × 10^9^ vg	8 and 16 months	[199]
	MPS I	Mouse	C57BL/6-*Idua*^−/−^	AAV2 or AAV5-PGK-hIDUA	10^9^ vg	6–7 weeks	[200]
	MPS I	Mouse	C57BL/6-*Idua*^−/−^	AAV2 or AVA5-mPGK-hIDUA	10^9^ vg	6–7 weeks	[200]
	MPS IIIA	Mouse	C57BL/6-*Sgsh*^−/−^	AAVrh10-PGK-SGSH-IRES-SUMF1	7.5 × 10^9^ gc	5 weeks	[201]
	MPS IIIA	Mouse	C57BL/6-*Sgsh*^−/−^	AAVrh10-CMV-SGSH-IRES-SUMF1	7.5 × 10^9^ gc	5 weeks	[201]
	MPS IIIA	Mouse	C57BL/6-*Sgsh*^−/−^	AAVrh10-CAG-hSGSH	8.4 × 10^8^, 4.1 × 10^10^, 9 × 10^10^ vg	5–6 weeks	[202]
	MPS IIIB	Mouse	C57BL/6-*Naglu*^−/−^	AAV2 or AAV5-mPGK-hNAGLU	10^9^ vg	6 weeks	[203]
	MPS IIIB	Mouse	C57BL/6-*Naglu*^−/−^	AAV5-CAG-hNAGLU	1.8 × 10^10^ vp	P2–4	[204]
	MPS IIIC	Mouse	Hgsnat^−/−^	AAV2TT-CAG-hcoHGSNAT	5.2 × 10^9^ vg	8 weeks	[205]
	MPS VII	Mouse	B6.C-H-2^bm1^/byBirgus^mps^/+	AAV5-RSV-GUSB	3 × 10^9^ vg	6–8 weeks	[206]
	MPS VII	Mouse	C3H/HeOuJ-*Gusb*^mps-2J^	AAV1, AAV9 or AAVrh10-GUSB-hGUSB	1.2–1.3 × 10^10^ vg	>2 months	[207]
	NP-A	Mouse	C57BL/6-ASMKO^−/−^	AVV2-CMV-hASM	1.86 × 10^10^ gc	10 weeks	[208]
	GM1	Cat	fGM1	AAV1 or AAVrh8-CAG-fβgal	3 × 10^11^, 4 × 10^12^ or 1.2 × 10^13^ vg	1.3–3 months	[209]
	GM2 (Sandhoff)	Cat	fG_M2_	AAV1-CAG-fHEXA/B or AAV8-DC172-HEXA/B	3 × 10^11^–3.2 × 10^12^ vg	0.9–2.6 months	[210]
	GM2 (Sandhoff)	Cat	fG_M2_	AAVrh8-CAG-HEXA, AAVrh8-CAG-HEXB	1.6–4.5 × 10^11^ vg	1 month	[211]
	MPS I	Dog	*IDUA* ^−/−^	AAV5-PGK-hIDUA	5 × 10^11^–2.1 × 10^12^ vg	18–30 months	[212]
	MPS IIIB	Dog	Schipperke *Naglu*^−/−^	AAV5-PGK-hNAGLU	5 × 10^11^–2.1 × 10^12^ vg	18–30 months	[212]
	CLN2 Batten	Rat	Fisher 344	AAV2-CAG-TPP1	10^9^–10^10^ pu	NA	[213]
	CLN2 Batten	NHP	C. sabaeus	AAV2-CAG-TPP1	3.6 × 10^11^ pu	5–10 years	[213]
Intrathecal (IT)							
	Krabbe	Mouse	B6.CE-*Galc*^twi^/J	AAV9, AAVrh10 or AAVOlig001-CAGGS-mβgal	2 × 10^11^ vg	P10–11	[214]
	MPS I	Mouse	C57BL/6-*IDUA*^−/−^	AAV2-CMV-hIDUA	2 × 10^9^–4 × 10^10^ vp	2–4 months	[215]
	MPS VII	Mouse	*Gusb* ^−/−^	AAV2-CMV-mGUSB	1 × 10^11^ and 5 × 10^11^ vp	P3 and 7–13 weeks	[216]
	GM1	Cat	*βgal* ^−/−^	AAVrh10-βgal	1 × 10^12^ vg/kg	NA	[217]
	MPS I	Cat	*IDUA* ^−/−^	AVV9-IDUA	1 × 10^12^ gc/kg	NA	[218]
Cisterna Magna (CM)							
	MPS IIIA	Mouse	C57BL/6-*SGSH*^D31N^	AAV9-CAG-mSgsh	5 × 10^9^–5 × 10^10^ vg	2 months	[219]
	MPS IIIB	Mouse	C57BL/6-*Naglu*^−/−^	AVV9-CAG-mNaglu	3 × 10^10^ vg	2 months	[220]
	MPS IIID	Mouse	C57BL/6NTac-*Gns*^tm1e(EUCOMM)Hmgu^	AAV9-CAG-mGns	5 × 10^10^ vg	2 months	[221]
	Pompe	Mouse	6neo/6neo	AAV9 or AAVrh10-CAG-hGAA	10^11^ vg	1 month	[222]
	AMD	Cat	*MANB* ^−/−^	AAV1-GUSB-fMANB	1 × 10^13^ gc	4–6 weeks	[223]
	GM1	Cat	*βgal* ^−/−^	AAVrh10-βgal	1 × 10^12^ vg/kg	NA	[217]
	MPS I	Cat	*IDUA* ^−/−^	AAV9-CB-fIDUA, AAV9-CMV-fIDUA	10^12^ gc/kg	4–7 months	[224]
	MPS I	Dog	*IDUA* ^−/−^	AAV9-CAG-IDUA	10^12^ gc/kg	1 month	[225]
	MPS I	Dog	*IDUA* ^−/−^	AAV9-CAG-IDUA	10^11^–10^12^ gc/kg	1 month	[226]
	MPS VII	Dog	*GUSB* ^−/−^	AAV9 or AAVrh10-CAG-caGUSB	5 × 10^12^ gc/kg	3 weeks	[227]
Intracerebroventricular (ICV)							
	CLN6 Batten	Mouse	*Cln6* ^nclf^	AAV9-CMV-hCLN6	5 × 10^10^–5 × 10^11^ vg	P0	[228]
	CLN8 Batten	Mouse	*C57BL/6J-Cln8* ^mnd^	AAV9.pT-MecP2.CLN8	5 × 10^10^ vg	P0–P1	[229]
	Gaucher	Mouse	CD1-K14-ln/ln	AAV9-GUSB-hGBA1	5 × 10^10^ gc	E16 and P0	[146]
	GM1	Mouse	*βgal* ^−/−^	AAV1-CAG-mβgal	3.2 × 10^9^ vg	P0	[230]
	Krabbe	Mouse	C57BL/6-twi^W339X^ or C57BL/129SVJ/FVB/N-twi-trs	AAV1-CMV-mGALC	3 × 10^10^ vp	P0–P1	[197]
	MLD	Mouse	C57BL/6J-*Asra*^tm1Gie^	AAV1-CAG-ARSA	2 × 10^11^ vg	8–12 weeks	[231]
	MLD	Mouse	C57BL/6J-*Asra*^tm1Gie^	AAV1, 9-CAG-ARSA	1.1 × 10^11^–2.3 × 10^11^ vg	8 and 18 weeks	[232]
	MPS I	Mouse	C57BL/6-*IDUA*^−/−^	AAV8-CAG-hIDUA	2 × 10^10^ vg	P4-P6	[233]
	MPS I	Mouse	C57BL/6-*IDUA*^null^	AAV5-CBA-hIDUA	1 × 10^11^ vp	8–11 weeks	[234]
	MPS IIIA	Mouse	B6Cg-*Sgsh*^(mps3/PstJ)^	AAV5-CMV-SGSH-IRES-SUMF1	6 × 10^9^–3 × 10^10^ vp	P0	[235]
	MPS III A	Mouse	B6Cg-*Sgsh*^(mps3a/PstJ)^	AAV9-CMV-IDSspSGSH-IRES-SUMF1	5.4 × 10^12^ gc/kg	P60	[236]
	MPS VII	Mouse	C3H/HeOuJ-*Gusb*^mps-2J^	AVV1, 2, 5-GUSB-GUSB	1.8 × 10^10^ vg	P0	[237]
	MPS VII	Mouse	B6.C-H-2^bm1^/byBirgus^mps^/+	AAV4-RSV-hGUSB	1 × 10^10^ vg	6–8 weeks	[238]
	MSD	Mouse	C57B6/S129J-*Sumf*^−/−^	AAV4, 9-CMV-SUMF1	1.2 × 10^10^ vg	P1	[152]
	NP-C	Mouse	BALB/cNctr-*Npc1*^m1N^/J	AAV9-hSYN-hNPC1	4.6 × 10^9^ vg	P0	[239]
	GM1	Cat	*βgal* ^−/−^	AAVrh10-βgal	1 × 10^12^ vg/kg	NA	[217]
	CLN2 Batten	Dog	Dachshunds *TPP1*^null^	AAV2-CAG-caTPP1	1.6 × 10^7^ vg	3 months	[240]
	CLN5 Batten	Sheep	*CLN5* ^−/−^	AAV9-CBh-CLN5	4 × 10^12^ vg	2–3 months	[241]
Combination							
IC + IT	CLN1 Batten	Mouse	C57BL/6-*Ptt1*^−/−^	AAV9-CAG-hPTT1	1 × 10^12^ vg/ml	P1	[191]
IC + ICV	GM2 (Sandhoff)	Cat	fG_M2_	AAVrh8-CAG-HEXA/B	1.1 × 10^12^ vg	1.1–1.6 months	[242]
IC + ICV	GM2 (Sandhoff)	Cat	fG_M2_	AAVrh8-CAG-HEXA, AAVrh8-CAG-HEXB	IC: 1.6–4.5 × 10^11^ vg; ICV: 6.4 × 10^11^ vg	1 month	[211]
CM + ICV	CLN2 Batten	Dog	Dachshunds *TPP1*^null^	AAV2-CAG-caTPP1	1.6 × 10^7^ vg	3 months	[240]
IC + ICV	CLN5 Batten	Sheep	*CLN5* ^−/−^	AAV9-CBh-CLN5	3.1 × 10^12^ vg	2–3 months	[241]
IC + ICV	GM2 (Tay-Sachs)	Sheep	TSD	AAVrh8-CAG-HEXA, AAVrh8-CAG-HEXA/B	6.3 × 10^12^ vg, 4.2 × 10^12^–1.3 × 10^13^ vg	2–4 months	[243]

## Data Availability

Not applicable.

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
