# Peer review of "Gene Therapy for Lysosomal Storage Disorders: Ongoing Studies and Clinical Development"

_biomolecules, 2021, doi:10.3390/biom11040611_

Round 1

Reviewer 1 Report

The manuscript entitle×’: “Gene Therapy for Lysosomal Storage Disorders: Ongoing Studies and Clinical Development” by Massaro et al, is a very comprehensive, well written manuscript. It summarizes the ongoing gene therapy preclinical and clinical studies for lysosomal storage disorders and describes future therapeutic modalities. It is a timely manuscript which should be published.

I have one concern. While most of the manuscript is very straightforward, the “Gene-editing” section is very difficult to follow. I do not think potential readers of this manuscript know what “Non Homologous End Joining” or “Homology Directed Repair” are, what ZFNs mean and more. I would simplify this section and add illustrations of the action of ZFNs and CRISPR/Cas9.

Other comments

  1. Page 2: 2nd paragraph: mannose-6-phosphate, not mannose-6phosphate
  2. Page 2: In Gaucher disease, the targeted enzyme has mannose termini, not mannose-6-p. Mannose terminated enzyme recognizes the mannose receptor on macrophages.
  3. Page 2: The authors mention that: “In addition, repeated infusions of high concentrations of the recombinant enzyme can lead to the development of immune reactions to the protein requiring complex management (21), particularly in patients carrying null mutations (22). ERT is not administered to patients with biallelic null mutations, they are too severe. Patients who receive ERT have at least one missense mutation.
  4. Page 4: The authors mention that adenoviruses have ITRs. Explanation for ITRs appears in page 5 , 6th Move it to page 4.
  5. Page 8: the authors explain that: “In summary, compared with other HSC-GT using ubiquitous promoters, this example offers evidence for a strategy using myeloid specific promoter to drive transgene expression as an alternative option of 1) boosting expression of the transgene in microglia/macrophages, which would be especially useful for LSDs with extensive CNS involvement….” I am not sure how transducing HSCs allows expression in microglia. Microglia do not originate from HSCs, they represent a unique brain population of cells.
  6. “Page 9, 2.5: “This is the case of cystinosis, a disease is caused by mutations in…”. Omit “is”.
  7. Page 9, 2.6:” The majority of these studies utilised lentiviral vectors to transduce HSCs and showed therapeutic benefits, suggesting HSC-GT appears to be a promising approach to treat LSDs with CNS involvement.” Not clear how HSCs populate the brain.
  8. Page 15: “However, the required dose required to achieve sufficient and extensive CNS expression via intravenous administration is high and raises concerns around toxicity (155)”. Omit the first: “required”.
  9. Page 15: “high doses of vector (6.7x0113 - 3x1014 vg/kg) “. Correct to: 6.7x 1013 - 3x1014 vg/kg.
  10. Page 15: “as result of a T cell mediated immune response to high amounts of capsid antigen (157). Should be: as a result of a…..
  11. Page 16: “…diseases caused by deficiencies in secreted enzymes IV gene delivery has also been used to successfully treat lysosomal defects involving…”. Omit “IV”.
  12. Table 4: In the Gaucher mouse model, the genotype should be: GBAL4444P/L444P.
  13. Page 18: in table 4 it is not clear what does it mean: 3.6x109 gc (genome content). Is this the number of calculated genome molecules? The most accurate to use is plaque forming units (pfu), which can be measured, and each particle contains one genome.
  14. Page 21, Please give the full name of the abbreviation HNPs when using the first time.
  15. In general: A schematic presentation of all viral genomes mentioned in the manuscript, with the site of insertion of the foreign cDNA, is recommended.
  16. The sections on ZFNs and on Talens are complicated. I suggest to simplify them and to add an illustration.
  17. Explanation of the CRISPR/Cas9 is too complicated as well. The authors could omit the description on the original bacterial system and explain the editing system, based on expression of Cas9 and guide RNA. Also, they should emphasize that although CRISPR/Cas9 technology is adequate for changing a mutant sequence into a normal one, there are efforts aiming at using this system to introduce an expression cassette into another locus on the genome, which acts as a safe harbor.
  18. On page 27 the authors explain that:” Another similar approach using the albumin locus as a safe harbour has also been investigated for the treatment of Tay-Sachs and Sandhoff diseases… . The inserted gene was a modified human Hex m subunit (HEXM) which incorporates the sequence of both a and b subunits and forms a homodimer capable of degrading GM2 gangliosides (207(, )300(, (301)”. More accurately, HEXM is a protein containing the α-subunit active site, the highly stable β-subunit interface, and surface areas from both subunits, that bind the GM2 activator protein (Tropaket al: Construction of a hybrid beta-hexosaminidase subunit capable of forming stable homodimers that hydrolyze GM2 ganglioside in vivo. Molecular therapy. Methods & clinical development, 2016, 3, 15057).
  19. Page 28: Please, explain what is NSG.
  20. Page 28: Please, explain what is gRNA.
  21. The authors mention: “...A method to overcome this limitation has been developed where the editor is split in half and fused to a fast-splicing split intein”. Please explain what are inteins.

Author Response

We thank the reviewer for their comments. We have addressed each comment in the document attached, and amended the manuscript accordingly. 

Reviewer 2 Report

The work submitted by Massaro and co-workers represents a very complete review on the topic of gene therapy for lysosomal storage disorders. 

The review is arranged according to a structure that makes it easy to follow by potential readers with different levels of knowledge of the field. Also, the work integrates very well all the most recent studies reported in the field. 

The introductory section presents also a good introduction to the problem, providing a context on which the review evolves. The sections are well balanced and discussed the topics in detail. 

The main drawback in my view is the language, which needs to be improved throughout the manuscript. 

Also the study would benefit from a more focused/less general conclusions section, highlighting in more detail the overall message from such a comprehensive study.

Author Response

We thank the reviewer for their comments. The manuscript has been revised to correct any English language related issues. We have also reviewed and build upon the 'Conclusions' paragraph (p 29).